

# The infrared physics of bad theories

## Benjamin Assel[1⋆] and Stefano Cremonesi[2†]

**1** CERN, Theoretical Physics Department, CH-1211 Geneva 23, Switzerland
**2** Department of Mathematical Sciences, Durham University, Durham DH1 3LE, UK

⋆ benjamin.assel@gmail.com   † stefano.cremonesi@durham.ac.uk

## Abstract

We study the complete moduli space of vacua of 3d $\mathcal{N} = 4$ $U(N)$ SQCD theories with $N_f$ fundamentals, building on the algebraic description of the Coulomb branch, and deduce the low energy physics in any vacuum from the local geometry of the moduli space. We confirm previous claims for good and ugly SQCD theories, and show that bad theories flow to the same interacting fixed points as good theories with additional free twisted hypermultiplets. A Seiberg-like duality proposed for bad theories with $N \leq N_f \leq 2N-2$ is ruled out: the spaces of vacua of the putative dual theories are different. However such bad theories have a distinguished vacuum, which preserves all the global symmetries, whose infrared physics is that of the proposed dual. We finally explain previous results on sphere partition functions and elucidate the relation between the UV and IR $R$-symmetry in this symmetric vacuum.

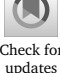
# 1 Introduction

The gauge coupling has positive mass dimension in three spacetime dimensions. This makes three-dimensional gauge theories super-renormalizable and free in the ultraviolet, regardless of the gauge group and matter content. At lower energies $\mu$, the dimensionless effective coupling $g_{\text{eff}}^2(\mu) = g^2(\mu)/\mu$ becomes stronger and interesting low energy physics can arise. Naively, the Maxwell/Yang-Mills term is irrelevant and drops out at low energies, leaving no mass scales. One might thus expect all 3d gauge theories to reach an interacting infrared fixed point. This is indeed the case if the number of matter fields $N_f$ is large: the gauge theory flows to a weakly coupled infrared fixed point in a large-$N_f$ expansion, with infrared effective coupling $g_{\text{eff}}^2 \sim 1/N_f$. This naive picture can be modified drastically by quantum effects. As the number of flavours $N_f$ is lowered, the infrared fixed point becomes more and more strongly coupled. Below a critical value $N_f = N_f^c$ for the number of flavours, however, a different low energy phase often kicks in, with spontaneous breaking of the flavour symmetry or a mass gap [1–4].

It is natural to ask whether the low energy phase diagram of three-dimensional gauge theories can be made more precise in the presence of supersymmetry. Our interest here is in 3d $\mathcal{N} = 4$ supersymmetric Yang-Mills theories (8 supercharges), which have low enough supersymmetry to allow matter fields but high enough to ensure theoretical control. We will focus for definiteness on 3d $\mathcal{N} = 4$ SQCD theories with $U(N)$ gauge groups and $N_f$ flavours of hypermultiplets in the fundamental representation.

A classification of 3d $\mathcal{N} = 4$ gauge theories according to their expected low energy properties was put forward by Gaiotto and Witten [5]. They assumed that a 3d $\mathcal{N} = 4$ gauge theory flows to a 3d $\mathcal{N} = 4$ SCFT in the infrared, and that the superconformal $R$-symmetry in the infrared is the same $R$-symmetry that is manifest at high energies. They then analysed whether this assumption is consistent with unitarity bounds applied to half-BPS gauge invariant chiral primary operators of an $\mathcal{N} = 2$ subalgebra. The bound $\Delta = R \geq 1/2$, where $\Delta$ is the conformal dimension, is automatically satisfied by operators built out of hypermultiplets. It is however non-trivial for 't Hooft monopole operators built out of vector multiplets, since their $R$-charges, which arise quantum-mechanically, are sensitive to the gauge group and matter content of the theory. In the terminology of [5], a 3d $\mathcal{N} = 4$ gauge theory is called *good* if all its monopole operators strictly obey the unitarity bound. A good theory is then expected to flow to an infrared SCFT with superconformal $R$-symmetry that is manifest in the UV. For $U(N)$ SQCD, this is the case if $N_f \geq 2N$. A gauge theory is instead called *ugly* if the unitarity bound is satisfied, but some monopole operators saturate it. It is then expected to flow to an IR SCFT whose superconformal $R$-symmetry is manifest in the UV, plus a decoupled free sector

given by the monopole operators which saturate the bound. For $U(N)$ SQCD, this happens if $N_f = 2N - 1$, and the monopole operators of magnetic charge $(\pm 1, 0, \ldots, 0)$ are the lowest components of the twisted hypermultiplet that becomes free in the infrared. Finally, a gauge theory is called *bad* if it has monopole operators with zero or negative $R$-charge. Because the naive unitarity bound is violated, a bad theory cannot flow to an SCFT whose superconformal $R$-symmetry is visible at high energies. For $U(N)$ SQCD, this happens if $N_f \leq 2N - 2$.

The infrared limit of bad theories is generally not well understood. It is expected that the monopole operators that violate the naive unitarity bound decouple at low energies, and that the leftover interacting part is described by an SCFT defined by a good theory, but the precise mechanism and the details are not clear. The intuition that the infrared SCFTs of good theories also describe the interacting infrared fixed points of bad theories is supported by the classification of dual $AdS_4$ type IIB backgrounds of [6, 7], which are in one-to-one correspondence with good linear and circular unitary quivers, leaving no room for holographic duals of bad quiver theories of these types. In the case of $U(N)$ SQCD theories, a concrete proposal for the infrared limit of a subclass of bad theories was made by Yaakov [8], based on mathematical identities [9] between the matrix integrals that calculate (regularized) partition functions on $S^3$ [10]. Yaakov conjectured that a *bad* SQCD theory with $U(N)$ gauge group and $N \leq N_f \leq 2N - 2$ flavours is infrared dual to the *good* $U(N_f - N)$ SQCD with $N_f$ flavours, plus $2N - N_f$ free twisted hypermultiplets. This generalizes the analogous statement made in [5] for the *ugly* $U(N)$ SQCD with $2N - 1$ flavours, which is expected to be IR dual to the *good* $U(N - 1)$ SQCD with $2N - 1$ flavours plus a single free twisted hypermultiplet.

The main purpose of this paper is to revisit these proposals and clarify the infrared fate of 3d $\mathcal{N} = 4$ $U(N)$ SQCD theories with $N_f$ flavours. We will determine the low energy effective field theory as a function of $N$, $N_f$ and, crucially, the supersymmetric vacuum. Indeed, 3d $\mathcal{N} = 4$ gauge theories have a rich moduli space of supersymmetric vacua, consisting of a Higgs branch $\mathcal{H}$, a Coulomb branch $\mathcal{C}$, and mixed branches, and the low energy theory critically depends on the choice of vacuum. We will determine the low energy theory by analysing the local geometry of the moduli space (at fixed $N$ and $N_f$) near any chosen vacuum. At a smooth point of moduli space, the low energy physics is governed by a set of free fields, and the metric on the moduli space is locally flat. More interesting physics occurs at singular points of moduli space: the low energy theory contains an interacting SCFT with extra massless degrees of freedom, and the metric on the space of vacua becomes locally conical. One can therefore identify low energy theories that involve interacting SCFTs by looking at conical singularities of the moduli space of supersymmetric vacua.

To perform this analysis we cannot rely on metric information on the full moduli space of vacua, because the non-perturbative corrections to the hyperkähler metric on the Coulomb branch [11] are not known explicitly for 3d $\mathcal{N} = 4$ $U(N)$ SQCD theories.[1] We will instead describe the moduli space of supersymmetric vacua as a complex algebraic variety, building on recent advances in understanding Coulomb branches of 3d $\mathcal{N} = 4$ gauge theories [15–17]. We will unify the well-known description of the classically exact Higgs branch with the more recent description of the quantum corrected Coulomb branch [17], providing a complete picture of the moduli space of supersymmetric vacua of 3d $\mathcal{N} = 4$ $U(N)$ SQCD theories at the quantum level. We will determine the singularity structure of the Coulomb branch, which corresponds to intersections of Coulomb and Higgs branch factors of mixed branches.

As a complex algebraic variety, the moduli space of vacua of a 3d $\mathcal{N} = 4$ gauge theory is independent of the real gauge coupling [17] and hence renormalization group invariant. The algebraic analysis of the moduli space of vacua therefore gives us direct information about the low energy physics. The geometry of the moduli space of the gauge theory zoomed near a particular vacuum must reproduce the moduli space of vacua of the low energy theory that

---

[1]See [12–14] for some explicit results in $SU(2)$ and pure $SU(N)$ theories.

the gauge theory flows to in that vacuum. By analysing the local algebraic geometry of the moduli space of vacua, we will thus be able to identify the infrared effective theory of good, ugly and bad 3d $\mathcal{N} = 4$ $U(N)$ SQCD theories, for *any* vacuum.[2] If the vacuum corresponds to a singular point in moduli space, the infrared theory contains an interacting SCFT, along with a free sector if this singular point is part of a singular locus of positive dimension. The geometry transverse to the singular locus determines the interacting SCFT, while the geometry tangent to the singular locus determines the free sector. Our analysis of $U(N)$ SQCD confirms that the infrared physics at any singular point of its Coulomb branch is always given by the infrared fixed point of a good theory, plus a number of free twisted hypermultiplets. We find that the infrared physics at a generic point of the codimension $r$ singular locus of the Coulomb branch of $U(N)$ SQCD with $N_f$ flavours is the same as that of the good $U(r)$ SQCD theory with $N_f$ flavours at the origin of its moduli space, plus $N - r$ free twisted hypermultiplets. The details of the infrared theory are controlled by the gauge and global symmetry breaking pattern in the given vacuum, analogously to what happens in four dimensions [18].

These results apply equally to good, ugly and bad $U(N)$ SQCD theories. The only difference is in the maximum value of $r$, the highest codimension of a singular locus in the Coulomb branch, which is equal to $N$ for good theories and to $\lfloor N_f/2 \rfloor < N$ for ugly/bad theories. For good $U(N)$ SQCD theories (with $N_f \geq 2N$ flavours), the singular locus of highest codimension in the Coulomb branch is just the origin of the full moduli space, at which the Higgs and Coulomb branch meet: this becomes the conformal vacuum of the infrared SCFT. For ugly and bad theories the singular locus of highest codimension in the Coulomb branch, at which the Coulomb branch meets the full Higgs branch, has positive dimension and is part of a mixed branch. This is due to the incomplete Higgsing on the Higgs branch.

Having understood the singularity structure of the full moduli space of vacua of $U(N)$ SQCD theories and the low energy physics at any point in moduli space, we can revisit the infrared dualities proposed for ugly and bad theories. In the case of the ugly $U(N)$ SQCD with $N_f = 2N - 1$ flavours, we confirm that the theory is infrared dual to the good $U(N-1)$ SQCD with $2N - 1$ flavours plus a free twisted hypermultiplets, by showing that the moduli spaces of vacua of the proposed dual theories are the same algebraic varieties. Instead we find that the bad $U(N)$ SQCD with $N \leq N_f \leq 2N - 2$ flavours is *not* infrared dual to the good $U(N_f - N)$ SQCD theory with $N_f$ flavours plus $2N - N_f$ free twisted hypermultiplets: the moduli spaces of the putative dual theories are different algebraic varieties. In fact, the full moduli space of vacua of the good $U(N_f - N)$ SQCD with $N_f$ flavours can be embedded in the moduli space of vacua of the bad $U(N)$ SQCD with $N_f$ flavours, but the remaining Coulomb branch moduli of the bad theory do not factorize, and the Higgs branch of the bad theory also contains higher dimensional components.

We find instead that for $N \leq N_f \leq 2N - 2$ there is a *symmetric vacuum* at which the low energy effective theory coincides with the fixed point of the putative dual good $U(N)$ SQCD, plus $2N - N_f$ decoupled free twisted hypermultiplets.[3] This vacuum is not the most singular point in the Coulomb branch, and one can flow to higher rank SCFTs at more singular locations. At the most singular locus of the Coulomb branch, the infrared physics consists of the infrared fixed point of $U(\lfloor \frac{N_f}{2} \rfloor)$ with $N_f$ flavours plus $N - \lfloor \frac{N_f}{2} \rfloor$ free twisted hypermultiplets. Instead the symmetric vacuum is singled out because it preserves all the global symmetries. While the infrared physics (and the local geometry of the moduli space of vacua) in the vicinity of the symmetric vacuum is the same as that of the dual theory proposed in [8], this statement does

---

[2]Our analysis was inspired by the analysis of the moduli space of vacua of 4d $\mathcal{N} = 2$ $SU(N)$ SQCD performed in [18], but we study the set of algebraic equations that define the 3d Coulomb branch instead of the Seiberg-Witten curve of the 4d theory.

[3]The symmetric vacuum does not exist for $N_f < N$. For ugly theories it is mapped to the origin of the moduli space of the dual good theory and of the extra $\mathbb{C}^2$. For good theories the symmetric vacuum is the origin of the moduli space, which becomes the conformal vacuum in the infrared.

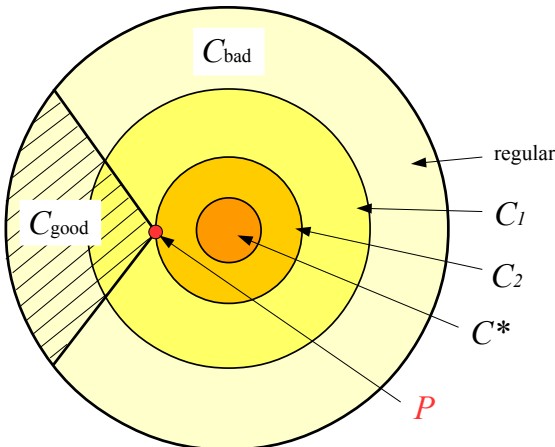

Figure 1: A schematic picture of the Coulomb branch of the bad theory $\mathscr{C}_{\mathrm{bad}}$ with its nested sequence of singular subloci $\mathscr{C}_1 \supset \mathscr{C}_2 \supset \cdots \supset \mathscr{C}^*$ of increasing codimension. The Coulomb branch of the good theory $\mathscr{C}_{\mathrm{good}}$ is included into $\mathscr{C}_{\mathrm{bad}}$ as a codimension $2N - N_f$ subvariety, and its most singular point $\mathscr{P}$ lies on a non-maximal singular subvariety of $\mathscr{C}_{\mathrm{bad}}$.

not extend globally. A schematic summary of these results is depicted in Figure 1. If a non-zero Fayet-Iliopoulos parameter is turned on, the Coulomb branch is lifted leaving only the symmetric vacuum, the Higgs branch is partially lifted and deformed to the cotangent bundle of the Grassmannian of $N$ planes in $N_f$ dimensions, and the moduli spaces of the supposedly dual theories match. This explains the relation between exact three-sphere partition functions, which are defined (by a suitable choice of contour integration) at non-zero FI parameter for bad theories. The picture that we have found is very reminiscent of that in 4d $\mathcal{N} = 2$ $SU(N)$ SQCD theories [18], which also fails to realize a Seiberg-like duality globally on the moduli space of vacua. The role of our symmetric vacuum is played there by the root of the baryonic branch.

Finally, we analysed how the twisted hypermultiplets that decouple at low energy at the symmetric vacuum transform under the $R$-symmetry of the UV and the IR SCFTs. The set of decoupling degrees of freedom always contains the chiral monopole operators of zero or negative UV $R$-charges, but $\mathcal{N} = 4$ supersymmetry requires that certain chiral monopole operators of positive UV $R$-charges pair up with those monopole operators to form free twisted hyper-multiplets. For bad theories, the $SU(2)_C$ $R$-symmetry which acts on the Coulomb branch and is manifest in the UV is unbroken in the symmetric vacuum, but it is different from the super-conformal $R$-symmetry of the infrared SCFT. The UV $SU(2)_C$ $R$-symmetry is instead a diagonal combination of the IR $SU(2)_C$ $R$-symmetry and of the principal embedding of $SU(2)$ inside the accidental flavour symmetry group $U(2N - N_f)$ acting on the free twisted hypermultiplets.

The rest of the paper is organized as follows. In Section 2 we discuss the moduli space of vacua of good SQCD theories, the structure of singularities and the low energy physics on the Coulomb branch. In Section 3 we analyse ugly theories, and in Section 4 we analyse bad theories. In Section 5 we elucidate the question of Seiberg duality, and show that there is no such duality for bad theories. We conclude with some future directions of research in Section 6. Some explicit examples of our general analysis are included in Appendix A.

## 2 The moduli space of vacua of good theories

In this section we study the space of vacua of $\mathcal{N} = 4$ $U(N)$ SQCD with $N_f \geq 2N$ flavours (fundamental hypermultiplets), which are *good* according to the classification of [5]. We start with a review of the classical moduli space of the theory, then we provide the description of the quantum Coulomb branch using the approach of [17] and we analyse its singularities, which we identify with roots of Higgs branches in the full moduli space of vacua. We also study the effect of massive deformations on the moduli space of vacua. Our results confirm previous statements in the literature.

### 2.1 Classical moduli space

The $U(N)$ SQCD theory has a vector multiplet with dynamical bosonic fields a gauge field $A_\mu$ and three real scalars $(\phi^1, \phi^2, \phi^3)$, valued in the $\mathfrak{u}(N)$ gauge algebra, and $N_f$ fundamental hypermultiplets whose bosonic fields are pairs of complex scalars $H_\alpha = (Q_\alpha, (\widetilde{Q}^\alpha)^\dagger)^T$, $\alpha = 1, \cdots, N_f$, transforming in the fundamental representation $\mathbf{N}$ of the gauge group.[4] Under the $R$-symmetry group $SU(2)_C \times SU(2)_H$, the vector multiplet scalars $\phi^i$ transform as a triplet of $SU(2)_C$ and the hypermultiplet scalars $(Q_\alpha, (\widetilde{Q}^\alpha)^\dagger)^T$ transform as a doublet of $SU(2)_H$. The hypermultiplet scalars can be assembled into an $N \times N_f$ complex matrix $Q = (Q^a{}_\alpha)$ and an $N_f \times N$ matrix $\widetilde{Q} = (\widetilde{Q}^\alpha{}_a)$, where $a = 1, \cdots, N$ is a colour index and $\alpha = 1, \cdots, N_f$ a flavour index.

The vacua of the theory are parametrized in part by the VEVs of vector multiplet and hypermultiplet scalars, which are constrained by the vacuum equations

$$
\begin{aligned}
\mu_{\hat{i}} &\equiv \mathrm{Tr}_2(HH^\dagger \sigma_{\hat{i}}) = 0 \,, & \hat{i} &= 1, 2, 3 \\
\epsilon_{ijk}[\phi^j, \phi^k] &= 0 \,, & i &= 1, 2, 3 \\
(\phi^i \otimes \mathbb{1}_2)H &= 0 \,, & i &= 1, 2, 3 \,.
\end{aligned}
\tag{2.1}
$$

Here indices $i, j, k$ label triplets of $SU(2)_C$, whereas $\hat{i}$ labels triplets of $SU(2)_H$. $\mathbb{1}_2$ is the identity matrix and $\sigma_{\hat{i}}$ are Pauli matrices, all acting on $SU(2)_H$ doublets, and $\mathrm{Tr}_2$ denotes the trace over $SU(2)_H$ doublet indices. Colour indices are not contracted in the first line of (2.1), which transforms in the adjoint representation of the gauge group, but flavour indices are contracted. One can turn on Fayet-Iliopoulos (FI) parameters, which are triplets of $SU(2)_H$ and would appear in the first line of (2.1), and also mass parameters, which are triplets of $SU(2)_C$ and would appear in the third line. We will briefly discuss their effect in Section 2.5.

The vacuum equations (2.1) can be obtained by dimensional reduction from $6d$ $\mathcal{N} = (1, 0)$ supersymmetry. The first line of (2.1) already appears in six dimensions and constrains Higgs branch components of the moduli space, where the hypermultiplet scalars take vacuum expectation value (VEV): it is an $SU(2)_H$ triplet of $D$-term equations, that sets to zero the moment maps of the $\mathfrak{u}(N)$ action on hypermultiplets. The remaining vacuum equations descend from gauge covariant kinetic terms in six dimensions, with $\phi^i = A_{3+i}$. The second line of (2.1) constrains Coulomb branch components of the moduli space, where vector multiplet scalars take VEV: it ensures that the adjoint scalars $\phi^i$ can be diagonalized simultaneously. Finally, the third line of (2.1) governs the interplay between Higgs and Coulomb branch factors of mixed branches of the moduli space of vacua.

More explicitly, for $N_f \geq 2N$ the last line of (2.1) implies that the classical moduli space of vacua $\mathcal{M}$ splits into the union of $N + 1$ subspaces

$$
\mathcal{B}_r = \mathcal{C}_r \times \mathcal{H}_{N-r} \,, \qquad r = 0, \cdots, N \,,
\tag{2.2}
$$

---

[4] $H_\alpha$ has an implicit colour index.

called *branches*, characterized by the fact that the vector multiplet scalars, which parametrize the Coulomb factor $\mathscr{C}_r$, take value in the Cartan subalgebra of a $\mathfrak{u}(r)$ subalgebra of $\mathfrak{u}(N)$,

$$\phi^i = \mathrm{diag}(\phi^i_1, \ldots, \phi^i_r, 0, \ldots, 0) \tag{2.3}$$

and the hypermultiplet scalars, which parametrize the Higgs factor $\mathscr{H}_{N-r}$, belong to the corresponding kernels,

$$Q = \begin{pmatrix} 0 \\ Q_{(N-r) \times N_f} \end{pmatrix}, \qquad \widetilde{Q}^\dagger = \begin{pmatrix} 0 \\ \widetilde{Q}^\dagger_{(N-r) \times N_f} \end{pmatrix}, \tag{2.4}$$

and have vanishing moment maps $\mu_{\hat{i}}$ for the $\mathfrak{u}(N-r)$ subalgebra of $\mathfrak{u}(N)$ that acts on their non-zero entries. At a generic point on $\mathscr{B}_r$ the gauge group is broken to $U(1)^r$.

The full classical moduli space has therefore the form

$$\mathcal{M} = \bigcup_{r=0}^{N} \mathscr{B}_r = \bigcup_{r=0}^{N} (\mathscr{C}_r \times \mathscr{H}_{N-r}), \tag{2.5}$$

with $\mathscr{C}_0 = \mathscr{H}_0 = \{0\}$ being a point. The top-dimensional Coulomb component $\mathscr{C}_N \equiv \mathscr{C}$ is called the *Coulomb branch*, and the top-dimensional Higgs component $\mathscr{H}_N \equiv \mathscr{H}$ is called the *Higgs branch*. With a slight abuse of notation, we will identify the branch $\mathscr{B}_N = \mathscr{C} \times \{0\}$ where the hypermultiplet scalars are all set to zero with the Coulomb branch $\mathscr{C}$, and the branch $\mathscr{B}_0 = \{0\} \times \mathscr{H}$ where the vector multiplet scalars are all set to zero with the Higgs branch $\mathscr{H}$. The other branches $\mathscr{B}_r$ with $r = 1, \ldots, N-1$ are called *mixed branches*.

Let us now describe the Higgs and Coulomb factors of the classical mixed branches in more detail, starting with the Higgs factors $\mathscr{H}_r$. The equations describing $\mathscr{H}_r$ are the same as those describing the Higgs branch of $U(r)$ SQCD with $N_f$ flavours, so it is enough to describe the Higgs branch $\mathscr{H} = \mathscr{H}_N$. The Higgs branch $\mathscr{H}$ of $U(N)$ SQCD with $N_f$ flavours is parametrized by the VEVs of the hypermultiplet scalars, subject to the triplet of $D$-term equations in the first line of (2.1) and quotiented by the gauge group action. This identifies the Higgs branch $\mathscr{H}$ with the hyperkähler quotient

$$\mathscr{H} = \vec{\mu}^{-1}(0)/U(N) = \mathbb{H}^{NN_f}////U(N), \tag{2.6}$$

which has quaternionic dimension $N(N_f - N)$. At a generic point on the Higgs branch $\mathscr{H}$, the gauge group is completely broken, the hypermultiplets are partially massive and the low energy physics is that of $N(N_f - N)$ free massless hypermultiplets. Importantly, the hyperkähler metric on $\mathscr{H}$ does not receive quantum corrections [19], therefore the classical description is exact.

For later purposes, it is useful to describe the moduli space of vacua as a complex algebraic variety in a fixed complex structure. This is equivalent to selecting an $\mathcal{N} = 2$ subalgebra of the $\mathcal{N} = 4$ superalgebra, with a manifest $R$-symmetry $U(1)_R \subset SU(2)_H \times SU(2)_C$. We choose the $U(1)_R$ symmetry which is the diagonal combination of the Cartan elements of $SU(2)_H \times SU(2)_C$. The hypermultiplets decompose into chiral multiplets $Q = (Q^a{}_a)$ and $\widetilde{Q} = (\widetilde{Q}^a{}_a)$ of $R$-charge $1/2$, which are subject to the $F$- and $D$-term equations

$$Q\widetilde{Q} = 0, \qquad QQ^\dagger - \widetilde{Q}^\dagger \widetilde{Q} = 0, \tag{2.7}$$

and to gauge equivalence. This describes the Higgs branch as a Kähler quotient,

$$\begin{aligned} \mathscr{H} &= \{Q \in \mathbb{C}^{N \times N_f}, \tilde{Q} \in \mathbb{C}^{N_f \times N} \mid Q\tilde{Q} = 0\}//U(N) \\ &\cong \{M \equiv \tilde{Q}Q \in \mathbb{C}^{N_f \times N_f} \mid M^2 = 0, \ \mathrm{rk}(M) \le N\}, \end{aligned} \tag{2.8}$$

where $\mathbb{C}^{a \times b}$ denotes the space of $a$-by-$b$ complex matrices. In the last expression we gave the equivalent description in terms of the gauge invariant meson operators $M$.

The classical description of the Coulomb factor $\mathscr{C}_r$ is the same as that of the Coulomb branch of $U(r)$ SQCD, so we can focus on the description of the Coulomb branch $\mathscr{C} = \mathscr{C}_N$ of $U(N)$ SQCD. The classical equations $[\phi^i, \phi^j] = 0$ imply that the matrices $\phi^i$ can be diagonalised simultaneously as in (2.3), leading to $3N$ real parameters $\phi_a^i$, $a = 1, \cdots, N$. In three dimensions, there are additional moduli related to the gauge field. This can be understood as follows. At a generic point on $\mathscr{C}$ the gauge group is broken to a maximal torus $U(1)^N$ by the $\phi^i$ VEVs. The abelian gauge connections $A_a$, $a = 1, \cdots, N$, for this abelian subgroup can be dualized via[5]

$$\frac{2\pi}{g^2} \star dA_a = i \, d\gamma_a \tag{2.9}$$

to periodic scalars $\gamma_a \sim \gamma_a + 2\pi$ called *dual photons*, which also take expectation value in the vacuum. Here $g$ is the bare Yang-Mills coupling. The naive Coulomb branch is therefore

$$\mathscr{C} \approx (\mathbb{R}^3 \times S^1)^N / S_N \,, \tag{2.10}$$

where the $\mathbb{R}^3$ factors are parametrized by $\phi_a^i$, the $S^1$ factors are parametrized by $\gamma_a$, and the quotient by the permutation group of $N$ elements $S_N$ arises from residual gauge transformations in the Weyl group. Formula (2.10) is usually referred to as the classical Coulomb branch. The approximate symbol means that this description only applies to generic points of the Coulomb branch, where all hypermultiplets are massive and the low energy physics is that of $N$ free abelian vector multiplets.

Due to $\mathcal{N} = 4$ supersymmetry, the Coulomb branch (as any of the $\mathscr{C}_r$) is a hyperkähler manifold with an $SU(2)$ isometry identified with the $SU(2)_C$ $R$-symmetry that acts on the vector multiplet scalars. Unlike the Higgs branch, the metric on the Coulomb branch receives quantum corrections (at one-loop for abelian theories and generically non-perturbatively) which affect the topology of the dual photon fibration. Taking into account the quantum corrections will lead to a globally consistent description of the full moduli space of vacua, preserving the mixed branch structure of equation (2.5).

## 2.2 The quantum Coulomb branch

In order to describe the exact Coulomb branch of the theory we will rely on the approach of [17], which proposes an algorithm to build the Coulomb branch as a complex algebraic variety with coordinates corresponding to VEVs of chiral monopole operators. To this end, let us first rewrite the *classical* Coulomb branch (2.10) as a complex algebraic variety in a fixed complex structure. The vector multiplet scalars $(\phi_a^i, \gamma_a)$ are rearranged into the complex scalars (which are lowest components of chiral superfields)

$$\varphi_a = \phi_a^1 + i\phi_a^2 \,, \qquad u_a^\pm = \exp\left[ \pm \left( \frac{2\pi}{g^2} \phi_a^3 + i\gamma_a \right) \right] . \tag{2.11}$$

$\varphi_a \in \mathbb{C}$ are the eigenvalues of the adjoint complex scalar $\Phi \equiv \phi^1 + i\phi^2$ of $R$-charge 1. The complex scalars $u_a^\pm$ satisfy the classical relations

$$u_a^+ u_a^- = 1 \qquad \text{(no sum over } a) \tag{2.12}$$

and parametrize $N$ copies of $\mathbb{C}^*$. The classical Coulomb branch is thus reexpressed as

$$\mathscr{C} \approx (\mathbb{C} \times \mathbb{C}^*)^N / S_N \tag{2.13}$$

---

[5]This formula holds in Euclidean signature. In Lorentzian signature there is no $i$ in the RHS.

and is parametrized by VEVs of symmetric (Weyl invariant) polynomials of $\varphi_a$ and $u_a^\pm$.

The single-valued operators $u_a^\pm$ are (bare) chiral 't Hooft monopole operators. Indeed, inserting the operator $(u_a^+)^{n_a}(x)$ if $n_a > 0$ (or $(u_a^-)^{-n_a}(x)$ if $n_a < 0$) in the Euclidean path integral is equivalent, by the duality (2.9), to requiring that gauge field configurations have a Dirac monopole singularity at the insertion point $x$ with flux[6]

$$\frac{1}{2\pi} \oint_{S_x^2} dA^{(a)} = n_a \in \mathbb{Z}. \tag{2.14}$$

A corresponding singularity is prescribed for $\phi_a^3$ in order to preserve half of the supersymmetry and therefore define a chiral operator for the fixed $\mathcal{N} = 2$ superalgebra. These bare monopole operators can further be dressed by the complex scalars $\varphi_a$. The symmetric polynomials in $\varphi_a$ and $u_a^\pm$ are thus gauge invariant dressed monopole operators.

The definition of monopole operators as singular boundary conditions in the path integral is better suited to the quantum theory, since it holds everywhere in moduli space, including points with enhanced gauge symmetry where the abelian duality breaks down. $\mathcal{N} = 4$ supersymmetry forbids dressed monopole operators to have superpotential constraints, but the operators still obey chiral ring relations that arise from the quantum dynamics of the theory [20] and translate into polynomial relations for the coordinates on $\mathscr{C}$. It is not straightforward to derive these relations.

To discuss the Coulomb branch of vacua and the associated chiral ring, it was however argued in [17] that it is sufficient to use the *abelianized* description in terms of Weyl invariant polynomials of $\varphi_a$ and $u_a^\pm$. This description is valid in a dense open subset of the Coulomb branch where the gauge group is broken to its maximal torus $U(1)^N \equiv \prod_{a=1}^N U(1)_a$ and all $W$-bosons have non-zero complex masses. In the quantum theory, the dependence of the bare monopole operators $u_a^\pm$ on the real scalars $\phi_a^3$ in the right of (2.11) receives quantum corrections and consequently the chiral ring relations (2.12) of the abelianized theory are modified. It was proposed in [17] that the quantum corrected abelianized relations that replace (2.12) are[7]

$$u_a^+ u_a^- \prod_{b \neq a} (\varphi_a - \varphi_b)^2 = \varphi_a^{N_f}, \qquad a = 1, \cdots, N. \tag{2.15}$$

The Coulomb branch (CB) relations of the non-abelian theory are obtained by recasting the relations (2.15) in terms of operators of the non-abelian theory, using the so-called *abelianization map* which expresses the VEV of any non-abelian dressed monopole operator as a Weyl invariant polynomial of the $u_a^\pm$ and $\varphi_a$. The Coulomb branch is generated by the subset of monopole operators of magnetic charge $(0, 0, \cdots, 0)$ and $(\pm 1, 0, \cdots, 0)$ [15, 17]. This means that the theory has infinitely many quantum relations, which allow to solve for all the other monopole operators in term of this finite basis, leaving only a finite number of relations between those. The (VEV of) operators in this basis are given in terms of the (VEV of) abelian operators by

$$
\begin{aligned}
\Phi_n &= \sum_{a_1 < \cdots < a_n} \varphi_{a_1} \cdots \varphi_{a_n} && (n = 1, \cdots, N) \\
V_n^\pm &= \sum_{a=1}^N u_a^\pm \sum_{\substack{b_1 < \cdots < b_n \\ b_i \neq a}} \varphi_{b_1} \cdots \varphi_{b_n} && (n = 0, \cdots, N-1).
\end{aligned} \tag{2.16}
$$

---

[6]For a generic gauge group $G$, the monopole charges are labelled by embeddings $U(1) \to G$ and $\vec{n}$ takes value in the coweight lattice of $G$, quotiented by the Weyl group.

[7]In [17] the factor $\prod_{b \neq a}(\varphi_b - \varphi_a)^2$ appears in the right hand side of the relation in the denominator, however it is implicitly assumed there that the relation can be brought to the above form and is still valid when two $\varphi_c$ VEVs coincide. Our sign conventions slightly differ from [17].

The CB relations of the non-abelian theory are then succinctly described by the generating polynomial relation

$$\mathcal{R}(z) := Q(z)\widetilde{Q}(z) + U^+(z)U^-(z) - P(z) = 0 \qquad \forall z \in \mathbb{C}, \tag{2.17}$$

with

$$Q(z) = \prod_{a=1}^{N}(z - \varphi_a), \quad U^{\pm}(z) = \sum_{a=1}^{N} u_a^{\pm} \prod_{b \neq a}(z - \varphi_b), \quad P(z) = z^{N_f}, \tag{2.18}$$

and $\widetilde{Q}$ is an auxiliary polynomial in $z$ of degree $\widetilde{N} = N_f - N$.[8] Thus $\mathcal{R}$ is a polynomial of degree $N + \widetilde{N} = N_f$. The map from the gauge invariant relations (2.17) to the abelianized relations (2.15) is obtained by evaluating the polynomial relation (2.17) at $z = \varphi_a$, $a = 1, \cdots, N$. The gauge invariant Coulomb branch (CB) operators $\Phi_n, V_n^{\pm}$ are identified with the coefficients of the polynomials $Q$ and $U^{\pm}$,

$$Q(z) = \sum_{n=0}^{N}(-1)^n \Phi_n z^{N-n}, \quad \widetilde{Q}(z) = \sum_{n=0}^{\widetilde{N}}(-1)^n \widetilde{\Phi}_n z^{\widetilde{N}-n},$$

$$U^{\pm}(z) = \sum_{n=0}^{N-1}(-1)^n V_n^{\pm} z^{N-1-n}, \tag{2.19}$$

with $\Phi_0 = 1$. The CB relations are obtained by setting to zero the coefficients $R_k$ of the polynomial $\mathcal{R}(z)$

$$\mathcal{R}(z) = \sum_{k=0}^{N_f}(-1)^k R_k z^{N_f - k}, \tag{2.20}$$

leading to $N_f + 1$ relations among the CB operators:

$$R_k := \sum_{n_1 + n_2 = k} \Phi_{n_1} \widetilde{\Phi}_{n_2} + \sum_{n_1 + n_2 = 2N - 2 - N_f + k} V_{n_1}^+ V_{n_2}^- - \delta_{k,0} = 0, \qquad (0 \leq k \leq N_f) \tag{2.21}$$

where the sum is over non-negative integers $n_1, n_2$.[9] The first $N_f - N + 1$ such relations determine the coefficients $\widetilde{\Phi}_n$ of $\widetilde{Q}$. The remaining $N$ relations are the non-trivial Coulomb branch relations of the non-abelian theory among the $2N$ dressed monopole operators $V_{0 \leq n \leq N-1}^{\pm}$ and the $N$ symmetric polynomials $\Phi_{1 \leq n \leq N}$, which were predicted using Hilbert series techniques in [15].[10] In the following we will find it convenient to view the coefficients of $\widetilde{Q}$ as additional CB operators and to manipulate the $N_f + 1$ relations altogether.

Note that the CB relations (2.21) are invariant under a $\mathbb{C}^*$ action (the complexification of the $U(1)_R$-symmetry) with charges

$$R[\Phi_n] = n, \qquad R[\widetilde{\Phi}_n] = n, \qquad R[V_n^{\pm}] = \frac{N_f}{2} - N + 1 + n. \tag{2.22}$$

The charges are all positive and the Coulomb branch is algebraically a cone.

So far we have only reviewed the context and gathered the ingredients necessary to start our analysis. We will next use this algebraic description to study the singularities of the Coulomb branch.

---

[8]The polynomials $Q(z)$ and $\tilde{Q}(z)$ are not to be confused with the hypermultiplet scalars which we denoted by the same letters. We hope that the distinction will be clear from the context.

[9]We have absorbed an inconsequential $(-1)^{N_f}$ factor in front of the monopole terms to simplify equations in the following.

[10]The Hilbert series technique only applies to good and ugly theories.

## 2.3 Singular loci and infrared SCFTs

The Coulomb branch geometry is singular along positive (quaternionic) codimension loci, signalling the presence of massless W-bosons and matter hypermultiplets, and the opening of Higgs branches. In this section we exhibit the nested structure of the Coulomb branch singular locus, with singular subspaces of increasing codimension.

The singular locus of the Coulomb branch $\mathscr{C}^{(1)}_{\rm sing}$ is described as the subvariety of $\mathscr{C}$ where the Jacobian matrix of the system of equations (2.21) degenerates, namely when its rank is not maximal. The Jacobian matrix $J = (J^i{}_k) = (\partial R_k / \partial \mathscr{O}_i)$ can be computed by differentiating the relations (2.21),

$$\sum_{n_1+n_2=k} (\Phi_{n_1} d\widetilde{\Phi}_{n_2} + \widetilde{\Phi}_{n_1} d\Phi_{n_2}) + \sum_{n_1+n_2=2N-2-N_f+k} (V^+_{n_1} dV^-_{n_2} + V^-_{n_1} dV^+_{n_2}) \equiv J^i{}_k d\mathscr{O}_i, \qquad (2.23)$$

with $0 \leq k \leq N_f$ and where $\mathscr{O}_i = (\Phi_n|\widetilde{\Phi}_m|V^+_p|V^-_q)$ collectively denote the coordinates on $\mathscr{C}$. We obtain

$$J =$$

$$\begin{pmatrix} 0 & & & 1 & & & & & & \\ \widetilde{\Phi}_0 & 0 & & \Phi_1 & 1 & & & & & \\ \vdots & \ddots & & \vdots & \ddots & & & & & \\ \widetilde{\Phi}_{\widetilde{N}-N+1} & & & \Phi_{\widetilde{N}-N+2} & & & V^-_0 & & V^+_0 & \\ \vdots & & & \vdots & & & & & & \\ \widetilde{\Phi}_{N-1} & \cdots & \widetilde{\Phi}_0 & \Phi_N & & & \vdots & \ddots & \vdots & \ddots \\ \vdots & & \vdots & & \ddots & & & & & \\ \widetilde{\Phi}_{\widetilde{N}-1} & \cdots & \widetilde{\Phi}_{\widetilde{N}-N} & & \Phi_N \cdots \Phi_1 & 1 & & & & \\ \widetilde{\Phi}_{\widetilde{N}} & \cdots & \widetilde{\Phi}_{\widetilde{N}-N+1} & & \ddots & \Phi_1 & V^-_{N-1} \cdots V^-_0 & & V^+_{N-1} \cdots V^+_0 & \\ & \ddots & \vdots & & & \vdots & & \ddots & \vdots & \ddots & \vdots \\ & & \widetilde{\Phi}_{\widetilde{N}} & & & \Phi_N & & V^-_{N-1} & & V^+_{N-1} \end{pmatrix}, \qquad (2.24)$$

where we have only indicated non-zero entries. The singular locus corresponds to the points where the above matrix has rank smaller than $N_f + 1$ and which belong to the Coulomb branch, that is satisfying (2.21).

There is an obvious singular locus given by

$$\mathscr{C}^{(1)}_{\rm sing} = \{\Phi_N = \widetilde{\Phi}_{\widetilde{N}} = V^+_{N-1} = V^-_{N-1} = 0\} \cap \mathscr{C}. \qquad (2.25)$$

In this case the rank of $J$ is reduced because the last row vanishes. This subvariety is described by the equations

$$\sum_{n_1+n_2=k} \Phi_{n_1} \widetilde{\Phi}_{n_2} + \sum_{n_1+n_2=2N-2-N_f+k} V^+_{n_1} V^-_{n_2} = \delta_{0,k}, \quad 0 \leq k \leq N_f - 2, \qquad (2.26)$$

where only the operators $\Phi_{1 \leq n \leq N-1}$, $\widetilde{\Phi}_{0 \leq n \leq N_f - N - 1}$ and $V^\pm_{0 \leq n \leq N-2}$ appear. This is isomorphic to the Coulomb branch of the good SQCD theory with gauge group $U(N-1)$ and $N_f - 2$ flavours:

$$\mathscr{C}^{(1)}_{\rm sing} \cong \mathscr{C}_{U(N-1),N_f-2}, \qquad (2.27)$$

where we introduced the notation $\mathscr{C}_{U(p),q}$ for the Coulomb branch of $U(p)$ SQCD with $q$ fundamental flavours.[11]

---

[11]In this notation, $\mathscr{C} = \mathscr{C}_{U(N),N_f}$.

This is the physically expected result: the singular space corresponds to having a triple $(\varphi_a, u_a^+, u_a^-)$ vanishing, giving rise to massless hypermultiplets.[12] We checked that there are no other singular loci for $N = 2, 3$ (and any $N_f$). For arbitrary value of $N$ it becomes more difficult to show mathematically that the Coulomb branch has no other singular submanifold, however this is still the physically expected result.

Since the singular locus $\mathscr{C}_{\text{sing}}^{(1)}$ is isomorphic to the Coulomb branch $\mathscr{C}_{U(N-1), N_f-2}$, it contains itself a singular subvariety $\mathscr{C}_{\text{sing}}^{(2)}$. Proceeding recursively we find a nested sequence of singular loci $\mathscr{C}_{\text{sing}}^{(r)}$, $1 \leq r \leq N$, isomorphic to the Coulomb branch of the $U(N-r)$ theory with $N_f - 2r$ flavours,

$$\mathscr{C}^* \equiv \mathscr{C}_{\text{sing}}^{(N)} \subset \cdots \subset \mathscr{C}_{\text{sing}}^{(r)} \subset \mathscr{C}_{\text{sing}}^{(r-1)} \subset \cdots \subset \mathscr{C}_{\text{sing}}^{(0)} \equiv \mathscr{C},$$

$$\mathscr{C}_{\text{sing}}^{(r)} = \{\Phi_{N-i} = \widetilde{\widetilde{\Phi}}_{\widetilde{N}-i} = V_{N-1-i}^+ = V_{N-1-i}^- = 0 \,|\, i = 0, \cdots, r-1\} \cap \mathscr{C} \tag{2.28}$$

$$\cong \mathscr{C}_{U(N-r), N_f-2r},$$

with $\mathscr{C}_{\text{sing}}^{(0)} = \mathscr{C}$ the full Coulomb branch. The singular subvariety $\mathscr{C}_{\text{sing}}^{(r)}$ is the locus in the Coulomb branch $\mathscr{C}$ where the Jacobian matrix has rank reduced by $r$ at least. $r$ is also the quaternionic codimension of $\mathscr{C}_{\text{sing}}^{(r)}$ inside $\mathscr{C}$; we will refer to it simply as the codimension in the following. The singular locus of highest codimension, the *most singular locus* $\mathscr{C}^*$, is reached for $r = N$ and contains a single point of the full Coulomb branch, the *origin of* $\mathscr{C}$,

Good theories $(N_f \geq 2N)$: $\qquad \mathscr{C}^* = \{\Phi_{n>0} = 0, \, \widetilde{\Phi}_0 = 1, \, \widetilde{\Phi}_{n>0} = 0, \, V_n^\pm = 0\}. \tag{2.29}$

In order to understand the infrared physics when sitting at a given point on a singular submanifold, we must study the geometry close to this singular point, which is identified with the Coulomb branch of the infrared theory. First we remark that the geometry close to the origin $\mathscr{C}^*$ is isomorphic to the full Coulomb branch $\mathscr{C}$: indeed the Coulomb branch of a good theory is algebraically a cone, invariant under rescaling (a $\mathbb{C}^*$ action with positive weights), and $\mathscr{C}^*$ is the tip of this cone, the fixed point of the $\mathbb{C}^*$ action. This signals the presence of an interacting CFT, which we denote $T_{U(N), N_f}$.[13]

Close to a generic point of $\mathscr{C}_{\text{sing}}^{(r)}$, namely away from the higher codimension subspace $\mathscr{C}_{\text{sing}}^{(r+1)}$, the local geometry $\mathscr{U}[\mathscr{C}_{\text{sing}}^{(r)}]$ of the Coulomb branch is described by taking a certain limit of the CB relations. The most direct way to study the local geometry for $r > 0$ is from the abelianized relations (2.15). The codimension $r$ singular locus $\mathscr{C}_{\text{sing}}^{(r)}$ is characterized by the vanishing of the operators $\Phi_{N-i} = \widetilde{\Phi}_{\widetilde{N}-i} = V_{N-1-i}^\pm = 0$ for $i = 0, \cdots, r-1$. This corresponds to having $r$ vanishing triples $(u_a^+, u_a^-, \varphi_a)$ out of $N$. Let us assume without loss of generality that this happens for $a = 1, \cdots, r$. We then take the limit $|u_a^\pm|, |\varphi_a| \ll 1$ for $1 \leq a \leq r$, keeping $u_a^\pm, \varphi_a$ of order one for $r + 1 \leq a \leq N$, in the abelianized relations (2.15). This leads to

$$u_a^+ u_a^- \prod_{b=r+1}^{N} \varphi_b^2 \prod_{\substack{b=1 \\ b \neq a}}^{r} (\varphi_a - \varphi_b)^2 = \varphi_a^{N_f}, \qquad a = 1, \cdots, r.$$

$$\tag{2.30}$$

$$u_a^+ u_a^- \prod_{b=r+1}^{N} (\varphi_a - \varphi_b)^2 = \varphi_a^{N_f - 2r}, \qquad a = r + 1, \cdots, N.$$

---

[12]Note that the $U(N-1)$ and $SU(N_f-2)$ (appearing in (2.27)) are the unbroken gauge and flavour symmetry group on the Higgs branch where the massless hypermultiplets takes VEV.

[13]In the classification of linear quiver SCFTs of [5], $T_{U(N), N_f}$ corresponds to $T_{\hat{\rho}}^{\rho}[SU(N_f)]$, with $\rho = (1, 1, \cdots, 1)$ and $\hat{\rho} = (N_f - N, N)$.

Redefining the abelian monopole operators $u_a^\pm \to u_a^\pm / \prod_{b=r+1}^N \varphi_b$, the equations in the first line become the abelianized relations of a (good) $U(r)$ gauge theory with $N_f$ flavours, and the region we are probing is the origin of its Coulomb branch. The local geometry close to this origin is scale invariant and matches the full Coulomb branch $\mathscr{C}_{U(r),N_f}$. The equations in the second line reproduce the abelianized relations of a $U(N-r)$ theory with $N_f - 2r$ flavours, and we are probing the region away from its singular locus, where it is parametrized by $N-r$ free twisted hypermultiplets. Locally the geometry is then a flat $\mathbb{C}^{2(N-r)}$ space. We conclude that the geometry close to any generic point of $\mathscr{C}_{\text{sing}}^{(r)}$ can be described algebraically as the product

$$\mathscr{U}[\mathscr{C}_{\text{sing}}^{(r)}] = \mathscr{C}_{U(r),N_f} \times \mathbb{C}^{2(N-r)}. \tag{2.31}$$

It is more involved to derive this result directly from the gauge invariant description. We do it in some simple examples in Appendix A.

From the local geometry (2.31) and the discussion leading to it, one can deduce the low-energy physics of the SQCD theory at any point on the Coulomb branch in terms of free fields and the interacting SCFTs $T_{U(r),N_f}$, with $r = 1, \cdots, N$. When flowing above a generic point $\mathsf{P} \in \mathscr{C}_{\text{sing}}^{(r)}$, $0 \le r \le N-1$, the infrared effective theory probes the region close to $\mathsf{P}$, which is of the form (2.31). The first factor matches the Coulomb branch of the SCFT $T_{U(r),N_f}$.[14] The second factor corresponds to the VEVs of $N-r$ free twisted hypermultiplets. We thus find the low-energy theory

$$\mathsf{P} \in \mathscr{C}_{\text{sing}}^{(r)} \quad \xrightarrow{IR} \quad T_{U(r),N_f} + (N-r) \text{ free twisted hypermultiplets}. \tag{2.32}$$

For $r = 0$, namely at a generic point on the Coulomb branch, the low-energy theory is that of $N$ free twisted hypermultiplets, as expected. As one goes to more singular loci on the Coulomb branch, the low-energy theory contains fewer free fields and a CFT of increasing rank $r$, until one reaches the origin of the Coulomb branch where there are no free fields and the low-energy physics is that of the $T_{U(N),N_f}$ SCFT.

## 2.4 The total moduli space

We can now reach a complete description of the moduli space of vacua, including Coulomb, Higgs and mixed branches, by showing that the codimension $r$ singular locus in the Coulomb branch $\mathscr{C}_{\text{sing}}^{(r)}$ coincides with the root of a Higgs factor of a mixed branch. This Higgs factor has complex dimension $2r(N_f - r)$ and is isomorphic to the full Higgs branch of a $U(r)$ SQCD theory with $N_f$ flavours.[15]

As reviewed in Section 2.1, the Higgs branch of $U(N)$ SQCD with $N_f$ massless flavours is

$$\begin{aligned} \mathscr{H} \equiv \mathscr{H}_{U(N),N_f} &= \{Q \in \mathbb{C}^{N \times N_f}, \tilde{Q} \in \mathbb{C}^{N_f \times N} \mid Q\tilde{Q} = 0\}/GL(N,\mathbb{C}) \\ &\cong \{M \equiv \tilde{Q}Q \in \mathbb{C}^{N_f \times N_f} \mid M^2 = 0, \ \text{rk}(M) \le N\} \,. \end{aligned} \tag{2.33}$$

The first line of (2.33) expresses the Higgs branch in terms of the gauge variant quarks and antiquarks $Q$ and $\tilde{Q}$, subject to the $F$-term equation $Q\tilde{Q} = 0$ due to the adjoint $\Phi$ in the vector multiplet, and quotiented by the action of the complexified gauge group. The second line describes the Higgs branch in terms of the gauge invariant mesons $M$. The latter expression implies that the Higgs branch is the closure of the nilpotent orbit $\mathscr{O}_{(2^N, 1^{N_f-2N})}$[16] of the

---

[14]We will see below that a Higgs factor of a mixed branch, isomorphic to the Higgs branch of $U(r)$ SQCD with $N_f$ flavours emanates from this singular locus, and that the full moduli space of $T_{U(r),N_f}$ is reproduced.

[15]Our analysis in this section follows closely the analysis of the non-baryonic Higgs branches of $4d$ $\mathcal{N} = 2$ $SU(N)$ SQCD with $N_f$ fundamental flavours performed in [18].

[16]$(2^N, 1^{N_f-2N})$ is shorthand for $(\underbrace{2, 2, \ldots, 2}_{N \text{ times}}, \underbrace{1, 1, \ldots, 1}_{N_f - 2N \text{ times}})$.

complexified flavour group $SL(N_f, \mathbb{C})$, which is defined by the condition $\mathrm{rk}(M) = N$. The closure $\overline{\mathcal{O}}_{(2^N, 1^{N_f - 2N})}$ is defined by the condition $\mathrm{rk}(M) \leq N$ and is the union of all the suborbits $\mathcal{O}_{(2^r, 1^{N_f - 2r})}$ with $r \leq N$ (for $N_f \geq 2N$). Let us denote by $\mathcal{H}_r$ the subvariety of the Higgs branch $\mathcal{H}$ corresponding to $\overline{\mathcal{O}}_{(2^r, 1^{N_f - 2r})}$:

$$\mathcal{H}_r \cong \{M \in \mathbb{C}^{N_f \times N_f} \mid M^2 = 0, \ \mathrm{rk}(M) \leq r\} \equiv \overline{\mathcal{O}}_{(2^r, 1^{N_f - 2r})}. \tag{2.34}$$

Explicitly, on $\mathcal{H}_r$ the quark chiral superfields $Q$ and $\tilde{Q}$ can be written up to gauge and flavour rotations as

$$
Q = \begin{pmatrix}
0 & \kappa_1 & 0 & 0 & \ldots & 0 & 0 & 0 & \ldots & 0 \\
0 & 0 & 0 & \kappa_2 & \ldots & 0 & 0 & 0 & \ldots & 0 \\
\vdots & & & & \ddots & & & & & \vdots \\
0 & 0 & 0 & 0 & \ldots & 0 & \kappa_r & 0 & \ldots & 0 \\
0 & 0 & 0 & 0 & \ldots & 0 & 0 & 0 & \ldots & 0 \\
\vdots & & & & & & & & & \vdots \\
0 & 0 & 0 & 0 & \ldots & 0 & 0 & 0 & \ldots & 0
\end{pmatrix},
$$

$$
\tilde{Q}^T = \begin{pmatrix}
\kappa_1 & 0 & 0 & 0 & \ldots & 0 & 0 & 0 & \ldots & 0 \\
0 & 0 & \kappa_2 & 0 & \ldots & 0 & 0 & 0 & \ldots & 0 \\
\vdots & & & & \ddots & & & & & \vdots \\
0 & 0 & 0 & 0 & \ldots & \kappa_r & 0 & 0 & \ldots & 0 \\
0 & 0 & 0 & 0 & \ldots & 0 & 0 & 0 & \ldots & 0 \\
\vdots & & & & & & & & & \vdots \\
0 & 0 & 0 & 0 & \ldots & 0 & 0 & 0 & \ldots & 0
\end{pmatrix},
\tag{2.35}
$$

so that the nilpotent meson matrix takes the Jordan normal form

$$
M = \begin{pmatrix}
0 & \kappa_1^2 & 0 & 0 & \ldots & 0 & 0 & 0 & \ldots & 0 \\
0 & 0 & 0 & 0 & \ldots & 0 & 0 & 0 & \ldots & 0 \\
0 & 0 & 0 & \kappa_2^2 & \ldots & 0 & 0 & 0 & \ldots & 0 \\
0 & 0 & 0 & 0 & \ldots & 0 & 0 & 0 & \ldots & 0 \\
\vdots & & & & \ddots & & & & & \vdots \\
0 & 0 & 0 & 0 & \ldots & 0 & \kappa_r^2 & 0 & \ldots & 0 \\
0 & 0 & 0 & 0 & \ldots & 0 & 0 & 0 & \ldots & 0 \\
0 & 0 & 0 & 0 & \ldots & 0 & 0 & 0 & \ldots & 0 \\
\vdots & & & & & & & & \ddots & \vdots \\
0 & 0 & 0 & 0 & \ldots & 0 & 0 & 0 & \ldots & 0
\end{pmatrix},
\tag{2.36}
$$

with $r$ non-trivial two-by-two nilpotent Jordan blocks. It is clear from (2.35) that at a generic point of $\mathcal{H}_r$ the $U(N)$ gauge group is broken to a residual gauge group $G_r = U(N - r)$ and the flavour group $SU(N_f)$ is broken to a residual flavour group $F_r = SU(N_f - 2r)$. By the Higgsing analysis, its quaternionic dimension is given by the number of residual neutral hypermultiplets:

$$\dim_{\mathbb{H}} \mathcal{H}_r = N_f N - (N^2 - (N - r)^2) - (N - r)(N_f - 2r) = r(N_f - r), \tag{2.37}$$

in agreement with the dimension of $\overline{\mathcal{O}}_{(2^r, 1^{N_f - 2r})}$ that can be computed purely using group theory [21].

Note that $\mathcal{H}_r$ are the same Higgs factors which appeared in (2.5). In fact $\mathcal{H}_r$ are the singular subvarieties of the full Higgs branch, which correspond to the fixed loci of a $U(N - r)$ subgroup of the $U(N)$ gauge group in the hyperkähler quotient construction. As explained in



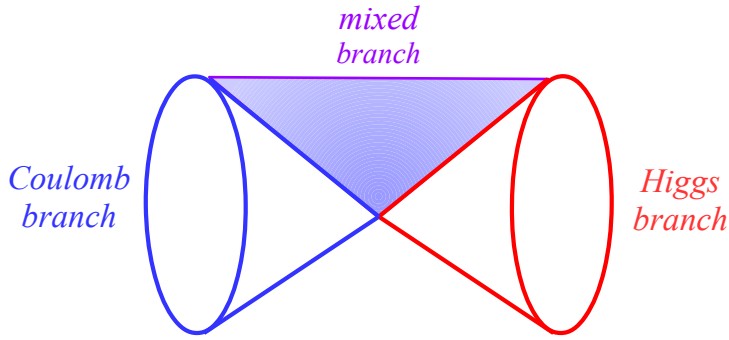

Figure 2: A schematic picture of the full moduli space of vacua, consisting of a Coulomb branch (blue), a Higgs branch (red) and a mixed branch (purple).

Section 2.1, the $r$-th Higgs factor $\mathscr{H}_r$ extends to a mixed Higgs-Coulomb branch, the Coulomb part of which corresponds to turning on the adjoint $\Phi$ and monopole operators for the unbroken $U(N-r)$ gauge group. Components of the Higgs branch with different $r$ extend differently to the Coulomb branch, and therefore should be treated as different: as $r$ decreases, the dimension of the Higgs factor of the mixed branch decreases, while the dimension of the Coulomb factor increases.

Taking into account the breaking of the gauge group to $U(N-r)$ and of the flavour group to $SU(N_f-2r)$ on $\mathscr{H}_r$, we conclude that the Higgs branch $\mathscr{H}_r$ is extended to a mixed branch $\mathscr{H}_r \times \mathscr{C}_{U(N-r),N_f-2r}$. The Coulomb branch factor of this mixed branch is nothing but the codimension $r$ singular locus $\mathscr{C}_{\text{sing}}^{(r)}$ of the full Coulomb branch (2.28). At the root of $\mathscr{H}_r$, which corresponds to $\mathscr{C}_{\text{sing}}^{(r)} \times \mathscr{H}^*$, with $\mathscr{H}^*$ the point where the mesons vanish ($M=0$), more transverse directions open up along the Coulomb branch and one is probing the region inside the full Coulomb branch around points in $\mathscr{C}_{\text{sing}}^{(r)}$. The local geometry of the Coulomb branch has the form of equation (2.31), where $\mathbb{C}^{2(N-r)}$ describes the tangent directions to $\mathscr{C}_{\text{sing}}^{(r)}$ at a generic point, and $\mathscr{C}_{U(r),N_f}$ describes the transverse geometry to the singularity in the full Coulomb branch. $U(r)$ is the gauge group which is broken on the Higgs branch factor $\mathscr{H}_r \cong \mathscr{H}_{U(r),N_f}$, and unbroken at its root. Taking into account mixed branches as well, it is easy to see that the local geometry $\mathscr{U}_{tot}[\mathscr{C}_{\text{sing}}^{(r)} \times \mathscr{H}^*]$ of the full moduli space of $U(N)$ SQCD with $N_f$ flavours near a generic point of $\mathscr{C}_{\text{sing}}^{(r)} \times \mathscr{H}^*$ is

$$\mathscr{U}_{tot}[\mathscr{C}_{\text{sing}}^{(r)} \times \mathscr{H}^*] \cong \mathbb{C}^{2(N-r)} \times \mathscr{M}_{U(r),N_f} \,, \tag{2.38}$$

where $\mathscr{M}_{U(r),N_f}$ denotes the full moduli space of $U(r)$ SQCD with $N_f$ flavours. The right-hand-side is precisely the moduli space of vacua of the IR effective theory at a generic point of $\mathscr{C}_{\text{sing}}^{(r)}$, in agreement with (2.32).

We see therefore that the structure of the full moduli space of vacua of $U(N)$ with $N_f \geq 2N$ flavours is the union of branches

$$\mathscr{M} = \bigcup_{r=0}^{N} (\mathscr{C}_{N-r} \times \mathscr{H}_r)\,, \tag{2.39}$$

with $\mathscr{H}_r \cong \mathscr{H}_{U(r),N_f}$ and $\mathscr{C}_{N-r} \equiv \mathscr{C}_{\text{sing}}^{(r)} \cong \mathscr{C}_{U(N-r),N_f-2r}$. A schematic picture with only three branches is shown in Figure 2.

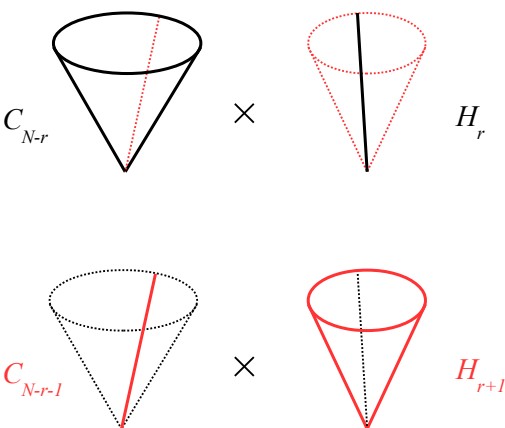

Figure 3: Two mixed branches $\mathscr{C}_{N-r} \times \mathscr{H}_r$ (black) and $\mathscr{C}_{N-r-1} \times \mathscr{H}_{r+1}$ (red) intersecting on the common subvariety $\mathscr{C}_{N-r-1} \times \mathscr{H}_r$.

Different branches intersect on singular subvarieties of the moduli space, with the branches opening in different directions, as schematically depicted in Figure 3,

$$(\mathscr{C}_{N-r_1} \times \mathscr{H}_{r_1}) \cap (\mathscr{C}_{N-r_2} \times \mathscr{H}_{r_2}) = \mathscr{C}_{N-r_{\max}} \times \mathscr{H}_{r_{\min}}, \qquad (2.40)$$

with $r_{\min} = \min(r_1, r_2)$ and $r_{\max} = \max(r_1, r_2)$. The Higgs branch is $\mathscr{H} \cong \mathscr{C}_0 \times \mathscr{H}_N = \mathscr{C}^* \times \mathscr{H}_N$. The Coulomb branch is $\mathscr{C} \cong \mathscr{C}_N \times \mathscr{H}_0 = \mathscr{C}_N \times \mathscr{H}^*$. The origin of the full moduli space is $\mathscr{C}^* \times \mathscr{H}^*$, the intersection point of all branches, where one flows to the $T_{U(N),N_f}$ SCFT at its conformal vacuum. These results are in agreement with [5]. The goal of this work is to perform a similar analysis for ugly and bad theories.

From the Higgs and Coulomb branch analysis of [5], it is expected that the previous picture generalizes to arbitrary good linear quiver theories with unitary gauge nodes, with the property that RG flows on their moduli space of vacua end in the class of SCFTs $T_{\hat\rho}^\rho[SU(N)]$ [5] and that the branches meet at a point corresponding to the most singular locus on the Coulomb branch.

## 2.5 Masses, FI parameters and moduli space of vacua

$U(N)$ SQCD theories with $N_f$ fundamental hypermultiplets have two kinds of relevant deformations compatible with $\mathcal{N} = 4$ supersymmetry: mass terms and FI terms.

Mass parameters are obtained by turning on constant commuting values for the $SU(2)_C$ triplets of scalars in the background vector multiplet for the $SU(N_f)$ flavour symmetry that acts on the Higgs branch. Choosing a complex structure, the triplet of masses decomposes into a complex and a real mass. We are interested in the complex structure, which is sensitive to the complex mass, but not to the real mass. Algebraically, the relevant parameters are therefore complex masses $m_\alpha$, with $\sum_{\alpha=1}^{N_f} m_\alpha = 0$,[17] which act as equivariant parameters for (the Cartan subalgebra of) the flavour symmetry $SU(N_f)$. The mass deformation generically lifts the Higgs branch and all mixed branches, except for the full Coulomb branch which is deformed. What remains of the Higgs branch is its origin $\mathscr{H}^*$, which is the fixed point of the action of the

---

[17]More precisely the sum $\sum_{\alpha=1}^{N_f} m_\alpha$ can be set to an arbitrary complex value by a common shift of all masses, which is unphysical.

flavour symmetry.[18] The Coulomb branch is not lifted, but its complex structure is deformed as follows [17]. The abelianized relations (2.15) become

$$u_a^+ u_a^- \prod_{b \neq a} (\varphi_a - \varphi_b)^2 = \prod_{\alpha=1}^{N_f} (\varphi_a - m_\alpha) \,, \qquad a = 1, \cdots, N \,, \tag{2.41}$$

where the right-hand-side is the product of the effective complex masses of the flavour hypermultiplets. The generating polynomial of chiral ring relations still has the form (2.17), but $P(z) = z^{N_f}$ is replaced by the characteristic polynomial of the $SU(N_f)$ flavour symmetry,

$$P(z) = \prod_{\alpha=1}^{N_f} (z - m_\alpha) = \sum_{n=0}^{N_f} (-1)^n M_n z^{N_f - n} \,, \tag{2.42}$$

where $M_n = \sum_{\alpha_1 < \cdots < \alpha_n} m_{\alpha_1} \cdots m_{\alpha_n}$ ($M_0$ is equal to 1 and $M_1$ can be set to 1 by shifting $z$). For generic values of the masses, the deformed Coulomb branch is non-singular.

On the other hand, Fayet-Iliopoulos parameters are obtained by turning on a constant value for the $SU(2)_H$ triplet of scalars in the background twisted vector multiplet for the $U(1)_J$ topological symmetry that acts on the Coulomb branch and assigns charges $\pm 1$ to the monopole operators $V_n^\pm$. Again, the complex structure of the moduli space of vacua is only affected by a complex FI parameter $\zeta$. A non-vanishing $\zeta$ gives mass to the dynamical vector multiplet and lifts the Coulomb branch. The only Coulomb vacuum that survives the deformation is the origin $\mathscr{C}^*$ of the undeformed Coulomb branch, which is the only vacuum invariant under the topological symmetry. The Higgs branch (2.33) is instead deformed by the FI parameter into a non-singular space. The $F$-term equation $Q\tilde{Q} = \zeta \mathbb{1}_N$ implies the gauge invariant relation $M^2 = \zeta M$. Up to gauge and flavour rotations, the hypermultiplet scalars take the form

$$
\begin{aligned}
Q &= \begin{pmatrix}
0 & \kappa_1 & 0 & 0 & \ldots & 0 & 0 & 0 & \ldots & 0 \\
0 & 0 & 0 & \kappa_2 & \ldots & 0 & 0 & 0 & \ldots & 0 \\
\vdots & & & & \ddots & & & & & \vdots \\
0 & 0 & 0 & 0 & \ldots & 0 & \kappa_N & 0 & \ldots & 0
\end{pmatrix}, \\
\tilde{Q}^T &= \begin{pmatrix}
\lambda_1 & \tilde{\kappa}_1 & 0 & 0 & \ldots & 0 & 0 & 0 & \ldots & 0 \\
0 & 0 & \lambda_2 & \tilde{\kappa}_2 & \ldots & 0 & 0 & 0 & \ldots & 0 \\
\vdots & & & & \ddots & & & & & \vdots \\
0 & 0 & 0 & 0 & \ldots & \lambda_N & \tilde{\kappa}_N & 0 & \ldots & 0
\end{pmatrix},
\end{aligned}
\tag{2.43}
$$

with $\kappa_a \tilde{\kappa}_a = \zeta$ for all $a = 1, \cdots, N$. The meson matrix takes the Jordan normal form

$$
M = \begin{pmatrix}
0 & \lambda_1 \kappa_1 & 0 & 0 & \ldots & 0 & 0 & 0 & \ldots & 0 \\
0 & \zeta & 0 & 0 & \ldots & 0 & 0 & 0 & \ldots & 0 \\
0 & 0 & 0 & \lambda_2 \kappa_2 & \ldots & 0 & 0 & 0 & \ldots & 0 \\
0 & 0 & 0 & \zeta & \ldots & 0 & 0 & 0 & \ldots & 0 \\
\vdots & & & & \ddots & & & & & \vdots \\
0 & 0 & 0 & 0 & \ldots & 0 & \lambda_N \kappa_N & 0 & \ldots & 0 \\
0 & 0 & 0 & 0 & \ldots & 0 & \zeta & 0 & \ldots & 0 \\
0 & 0 & 0 & 0 & \ldots & 0 & 0 & 0 & \ldots & 0 \\
\vdots & & & & & & & & \ddots & \vdots \\
0 & 0 & 0 & 0 & \ldots & 0 & 0 & 0 & \ldots & 0
\end{pmatrix}, \tag{2.44}
$$

---

[18]If non-generic masses associated to a subgroup $G_F' \subset SU(N_f)$ of the flavour symmetry are turned on, the set of fixed points of the action of $G_F'$ on the Higgs branch is not lifted.

with $N$ eigenvalues equal to $\zeta$ and $N_f - N$ eigenvalues equal to 0. Note that this is a deformation of the full Higgs branch $\mathscr{H}_N$ of quaternionic dimension $N(N_f - N)$,[19] that emanates from the origin $\mathscr{C}^*$ of the Coulomb branch in the absence of an FI parameter. (As a hyperkähler manifold, the deformed Higgs branch is the cotangent bundle over the Grassmannian $\mathrm{Gr}(N, N_f)$.) All the other mixed branches are lifted.

## 3 Ugly theories

Before moving to the study of bad theories we briefly address the question of *ugly* theories, corresponding to $N_f = 2N - 1$. It was found in [5] that the space of vacua has a branch $\mathbb{C}^2 \times \mathscr{H}_{U(N-1),2N-1}$, with $\mathbb{C}^2$ parametrizing the VEV of a free twisted hypermultiplet and $\mathscr{H}_{U(N-1),2N-1}$ isomorphic to the Higgs branch of $U(N-1)$ SQCD with $2N-1$ flavour hypermultiplets. It was deduced that at the origin of $\mathscr{H}_{U(N-1),2N-1}$ (and at any point along $\mathbb{C}^2$) the theory flows to the $T_{U(N-1),2N-1}$ SCFT with a decoupled free twisted hypermultiplet. This is referred to as an infrared duality between the ugly theory and the good $U(N-1)$ theory with $2N-1$ flavours and a decoupled twisted hypermultiplet. This duality was tested using sphere partition functions in [8, 22].

Here we confirm and complete these results by computing the Coulomb branch of the ugly theory and showing that it is *exactly* given by $\mathbb{C}^2 \times \mathscr{C}_{U(N-1),2N-1}$. One can show in general that the full moduli space of the ugly theory is of the form $\mathscr{M}_{\mathrm{ugly}} = \mathbb{C}^2 \times \mathscr{M}_{U(N-1),2N-1}$, the direct product of $\mathbb{C}^2$ with the full moduli space of the $U(N-1)$ good theory.[20] Therefore the duality between infrared SCFTs is corroborated by an exact agreement between the moduli space of vacua of the two dual theories at the level of the algebraic description, which is insensitive to the gauge coupling and therefore renormalization group invariant.

The Coulomb branch of the ugly theory with gauge group $U(N)$ and $N_f = 2N - 1$ fundamental hypermultiplets is described by the polynomial relation

$$Q(z)\widetilde{Q}(z) + U^+(z)U^-(z) = P(z),$$
$$\text{with} \quad [Q] = N, \quad [\widetilde{Q}] = N-1, \quad [U^\pm] = N-1, \quad [P] = 2N-1, \quad (3.1)$$

where $[X]$ denotes the degree of the polynomial $X$. The expansions of the polynomials are as follows

$$Q(z) = z^N - \sum_{n=0}^{N-1} (-1)^n \Phi_{n+1} z^{N-1-n}, \quad \widetilde{Q}(z) = z^{N-1} - \sum_{n=0}^{N-2} (-1)^n \widetilde{\Phi}_{n+1} z^{N-2-n},$$

$$U^\pm(z) = V_0^\pm z^{N-1} + \sum_{n=1}^{N-1} (-1)^n V_n^\pm z^{N-1-n}, \quad P(z) = z^{2N-1} + \sum_{n=1}^{2N-1} (-1)^n M_n z^{2N-1-n}. \quad (3.2)$$

Importantly $\widetilde{Q}(z)$ is a monic polynomial, *i.e.* its higher degree term $z^{N-1}$ has coefficient one,

---

[19]The deformed Higgs branch is a hyperkähler quotient of the baryonic branch of the $SU(N)$ SQCD theory with $N_f$ fundamentals studied in [18] by its baryonic $U(1)_B$ symmetry, with complex moment map equal to the complex FI parameter $\zeta$. The mesonic branch of [18] is lifted by the FI deformation.

[20]We leave the study of mixed branches as an exercise to the reader.

in order to solve (3.1). The relations (3.1) can be rearranged in the following dual form:[21]

$$Q_D(z)\widetilde{Q}_D(z) + U_D^+(z)U_D^-(z) = P(z),$$
$$\text{with} \quad [Q_D] = N-1, \quad [\widetilde{Q}_D] = N, \quad [U_D^\pm] = N-2, \quad [P] = 2N-1,$$
$$Q_D(z) = \widetilde{Q}(z),$$
$$\widetilde{Q}_D(z) = Q(z) - V_0^+ V_0^- \widetilde{Q}(z) + V_0^+ U^-(z) + V_0^- U^+(z),$$
$$U_D^\pm(z) = U^\pm(z) - V_0^\pm \widetilde{Q}(z).$$

(3.3)

This precisely describes the Coulomb branch of the good theory with $U(N-1)$ gauge group and $2N-1$ fundamental hypermultiplets. In addition we see that the monopole operators $V_0^\pm$, which carry $R$-charge $1/2$, decouple from the dual Coulomb branch equations (3.3) and yield a $\mathbb{C}^2$ factor corresponding to the VEV of a free twisted hypermultiplet.[22] The Coulomb branch of the ugly theory is therefore

$$\mathscr{C}_{\text{ugly}} := \mathscr{C}_{U(N),2N-1} = \mathbb{C}^2 \times \mathscr{C}_{U(N-1),2N-1}, \tag{3.4}$$

providing further support for the infrared duality with $U(N-1)$ SQCD with $2N-1$ flavours and a free twisted hypermultiplet.

# 4 Bad theories

We now reach the more interesting and rather unexplored territory of *bad* theories by considering the $U(N)$ SQCD theory with $N_f \leq 2N-2$ flavour hypermultiplets.

Bad theories have the distinctive feature that the gauge group cannot be completely higgsed, therefore all the branches of their moduli space have some Coulomb directions, and they admit monopole operators with negative or zero $U(1)_R$ $R$-charge. Since the $U(1)_R$ $R$-charge is identified with the conformal dimension for chiral operators in a super-conformal theory and this dimension cannot be smaller than one-half in a unitary CFT, it has been deduced that the $R$-symmetry of a candidate infrared SCFT cannot coincide with the UV $R$-symmetry. A possible scenario is that the negative (or zero) $R$-charge monopole operators decouple at low energy, becoming free twisted hypermultiplets with accidental symmetries, and that the $R$-symmetry of the infrared theory, which is made of an interacting SCFT and free twisted hypermultiplets, is a mixing of the UV $R$-symmetry and accidental global symmetries.[23] We will make this more precise at the end of Section 5.2. In this section we will study the full moduli space of vacua of the bad SQCD theories in detail and explain their infrared behaviour everywhere on this space.

The discussion of the space of vacua of bad SQCD theories is very similar to that of the good SQCD theories with a few crucial differences. The classical $D-$ and $F-$term equations are identical (given by (2.1)), but the solutions show only $\lfloor \frac{N_f}{2} \rfloor + 1$ branches $\mathscr{B}_r$, $r = 0, \cdots, \lfloor \frac{N_f}{2} \rfloor$, so the space of vacua has the form

$$\mathscr{M} = \bigcup_{r=0}^{\lfloor \frac{N_f}{2} \rfloor} \mathscr{B}_r = \bigcup_{r=0}^{\lfloor \frac{N_f}{2} \rfloor} (\mathscr{C}_{N-r} \times \mathscr{H}_r). \tag{4.1}$$

On the mixed branch with the smallest Coulomb factor $\mathscr{C}_{N-\lfloor N_f/2 \rfloor}$ and the largest Higgs factor $\mathscr{H}_{\lfloor N_f/2 \rfloor}$, $\lfloor N_f/2 \rfloor$ triples of scalars $(\varphi_a, u_a^+, u_a^-)$ vanish. $\mathscr{H}_{\lfloor N_f/2 \rfloor}$ is what we call the Higgs branch

---

[21]There is an ambiguity in the mapping of topological symmetries, corresponding to $U_D^+ \leftrightarrow U_D^-$.

[22]It is a twisted hypermultiplet since it is charged under the $U(1)_C \subset SU(2)_C$ $R$-symmetry.

[23]See also [23] on UV versus IR $R$-symmetries in bad theories.

$\mathcal{H}$. The Higgs factors $\mathcal{H}_r$ are described, as for good theories, as closures of nilpotent orbits of $SL(N_f, \mathbb{C})$, $\overline{\mathcal{O}}_{(2^r, 1^{N_f - 2r})}$, see (2.34).

We now consider the Coulomb branch of vacua $\mathscr{C} := \mathscr{C}_N$ using the formalism of [17], and analyse its singularity structure as we did for good theories in Section 2.

## 4.1 Coulomb branch

As for good theories, the algebraic description of the Coulomb branch $\mathscr{C}$ is based on the abelianized relations (2.15). The CB generators are again the complex scalar operators, $\Phi_n$, $1 \leq n \leq N$, and dressed monopole operators of magnetic charge $(\pm 1, 0, \cdots, 0)$, $V_n^{\pm}$, $0 \leq n \leq N - 1$. The CB relations are still captured by the polynomial relation (2.17), which we repeat here for convenience:

$$\mathcal{R}(z) := Q(z)\widetilde{Q}(z) + U^+(z)U^-(z) - P(z) = 0 \qquad \forall z, \tag{4.2}$$

where $Q$ and $U^{\pm}$ are polynomials whose coefficients coincide with the generators $\Phi_n$ and $V_n^{\pm}$, and $\widetilde{Q}$ is now an auxiliary polynomial of degree $\widetilde{N} = N - 2$:

$$Q(z) = \sum_{n=0}^{N} (-1)^n \Phi_n z^{N-n}, \quad \widetilde{Q}(z) = \sum_{n=0}^{N-2} (-1)^n \widetilde{\Phi}_n z^{\widetilde{N}-n},$$
$$U^{\pm}(z) = \sum_{n=0}^{N-1} (-1)^n V_n^{\pm} z^{N-1-n}, \quad P(z) = z^{N_f}, \tag{4.3}$$

where $\Phi_0 = 1$. The polynomial relation (4.2) is equivalent to the $2N - 1$ relations between the CB operators:

$$R_k := \sum_{n_1 + n_2 = k} \left( \Phi_{n_1} \widetilde{\Phi}_{n_2} + V_{n_1}^+ V_{n_2}^- \right) - (-1)^{N_f} \delta_{k, 2N - N_f - 2} = 0, \quad (0 \leq k \leq 2N - 2). \tag{4.4}$$

One can use the relations with $k = 0, \cdots, N - 2$ to solve for the $\widetilde{\Phi}_n$ in terms of the other generators, which are constrained by the $N$ remaining relations, however we find it again more convenient, and it will prove very useful, to keep the $\widetilde{\Phi}_n$ as additional generators and to work with the simple relations (4.4).

Note that the CB relations (4.4) are invariant under a $\mathbb{C}^*$ action (the complexification of the $U(1)_R$-symmetry) with charges

$$R[\Phi_n] = n, \qquad R[\widetilde{\Phi}_n] = N_f - 2N + 2 + n, \qquad R[V_n^{\pm}] = \frac{N_f}{2} - N + 1 + n. \tag{4.5}$$

Notice that there are monopole operators (and $\widetilde{\Phi}_n$ operators) with non-positive $R$-charge.

## 4.2 Singular loci and infrared SCFTs

The singular subspace of the Coulomb branch is obtained as in Section 2.3 by considering the Jacobian matrix of the system of equations (4.4) and looking for loci of reduced rank in the

Coulomb branch. The Jacobian matrix is

$$J^{(N,N_f)} =$$

$$\begin{pmatrix} 0 & & & & 1 & & & V_0^- & & & V_0^+ & & \\ \widetilde{\Phi}_0 & 0 & & & \Phi_1 & 1 & & V_1^- & V_0^- & & V_1^+ & V_0^+ & \\ \vdots & \ddots & \ddots & & \vdots & & \ddots & \vdots & & \ddots & \vdots & & \ddots \\ & & & & \Phi_{N-2} & \cdots & 1 & & & & & & \\ \widetilde{\Phi}_{N-2} & \cdots & \widetilde{\Phi}_0 & 0 & \Phi_{N-1} & \cdots & \Phi_1 & V_{N-1}^- & \cdots & V_0^- & V_{N-1}^+ & \cdots & V_0^+ \\ 0 & \widetilde{\Phi}_{N-2} & \cdots & \widetilde{\Phi}_0 & \Phi_N & \cdots & \Phi_2 & & \ddots & \vdots & & \ddots & \vdots \\ & \ddots & & \vdots & & \ddots & \vdots & & & & & & \\ & & & \widetilde{\Phi}_{N-2} & & & \Phi_N & & & V_{N-1}^- & & & V_{N-1}^+ \end{pmatrix} . \tag{4.6}$$

When $N_f \leq 1$ we find that there is no singularity on the Coulomb branch, which is therefore a smooth space. When $2 \leq N_f \leq 2N - 2$ we find that $J^{(N,N_f)}$ has reduced rank on the Coulomb branch locus $\Phi_N = \widetilde{\Phi}_{N-2} = V_{N-1}^\pm = 0$ (this locus is not part of the CB when $N_f \leq 1$). This is again the physically expected result: the singular space corresponds to having a triple $(\varphi_a, u_a^+, u_a^-)$ vanishing, giving rise to massless hypermultiplets. We explicitly checked that there are no other singular loci only for $N = 2, 3$ (and any $N_f$), and expect that this holds generally on physical grounds.

The equations $\Phi_N = \widetilde{\Phi}_{N-2} = V_{N-1}^\pm = 0$ define the singular locus of quaternionic codimension one. On this subvariety of the Coulomb branch, the equations (4.4) reduce to the CB relations of the $U(N-1)$ theory with $N_f - 2$ flavour hypermultiplets, hence the singular locus is isomorphic to the Coulomb branch $\mathscr{C}_{U(N-1),N_f-2}$:

$$\mathscr{C}_{\text{sing}}^{(1)} = \{\Phi_N = \widetilde{\Phi}_{N-2} = V_{N-1}^+ = V_{N-1}^- = 0\} \cap \mathscr{C} \cong \mathscr{C}_{U(N-1),N_f-2} . \tag{4.7}$$

When $N_f \leq 3$, the subspace $\mathscr{C}_{\text{sing}}^{(1)} \simeq \mathscr{C}_{U(N-1),N_f-2}$ is smooth. When $N_f \geq 4$, this singular locus contains itself a singular subspace $\mathscr{C}_{\text{sing}}^{(2)}$ which corresponds to CB loci where the rank of the Jacobian matrix (4.6) is reduced by two. This is the subvariety of the CB defined by $\Phi_N = \widetilde{\Phi}_{N-2} = V_{N-1}^\pm = 0$ and $\Phi_{N-1} = \widetilde{\Phi}_{N-3} = V_{N-2}^\pm = 0$, which is isomorphic to the Coulomb branch $\mathscr{C}_{U(N-2),N_f-4}$.

This structure goes on, leading to a nested sequence of singular subspaces of increasing codimension $\mathscr{C}_{\text{sing}}^{(r)}$, $r = 1, \cdots, \lfloor \frac{N_f}{2} \rfloor$,

$$\mathscr{C}^* \equiv \mathscr{C}_{\text{sing}}^{(\lfloor N_f/2 \rfloor)} \subset \cdots \subset \mathscr{C}_{\text{sing}}^{(r)} \subset \mathscr{C}_{\text{sing}}^{(r-1)} \subset \cdots \subset \mathscr{C}_{\text{sing}}^{(0)} \equiv \mathscr{C} ,$$
$$\mathscr{C}_{\text{sing}}^{(r)} = \{\Phi_{N-i} = \widetilde{\Phi}_{N-2-i} = V_{N-1-i}^+ = V_{N-1-i}^- = 0 \,|\, i = 0, \cdots, r-1\} \cap \mathscr{C} \tag{4.8}$$
$$\cong \mathscr{C}_{U(N-r),N_f-2r} ,$$

with $\mathscr{C}_{\text{sing}}^{(0)} = \mathscr{C}$ the full Coulomb branch. The sequence terminates at the most singular locus $\mathscr{C}^*$, which is isomorphic to the Coulomb branch of the $U(N - \lfloor \frac{N_f}{2} \rfloor)$ with zero or one flavour hypermultiplet depending on whether $N_f$ is even or odd: $\mathscr{C}^* \cong \mathscr{C}_{N-\lfloor \frac{N_f}{2} \rfloor, N_f \bmod 2}$.

The singularity structure (4.8) is completely analogous to that of Coulomb branches of good theories (2.28), except that the nested sequence of singular loci terminates with the most singular locus $\mathscr{C}^*$ which has positive quaternionic dimension $N - \lfloor \frac{N_f}{2} \rfloor$, rather than being a point.

To understand the low energy physics at the singular loci, we again study the CB geometry close to the singularities. The analysis of Section 2.3 remains valid for bad theories: in a neighbourhood $\mathscr{U}[\mathscr{C}_{\text{sing}}^{(r)}]$ of a generic point in $\mathscr{C}_{\text{sing}}^{(r)}$, the geometry has the form of a direct product,

with a factor isomorphic to the Coulomb branch of a good $U(r)$ theory with $N_f$ flavours and a "free" factor $\mathbb{C}^{2(N-r)}$,

$$\mathscr{U}[\mathscr{C}_{\text{sing}}^{(r)}] = \mathscr{C}_{U(r),N_f} \times \mathbb{C}^{2(N-r)}. \tag{4.9}$$

The factor $\mathbb{C}^{2(N-r)}$ corresponds to the directions tangent to the singular locus $\mathscr{C}_{\text{sing}}^{(r)} \simeq \mathscr{C}_{U(N-r),N_f-2r}$ near a generic point in $\mathscr{C}_{\text{sing}}^{(r)}$, which are parametrized by $N-r$ free twisted hypermultiplets. The factor $\mathscr{C}_{U(r),N_f}$ is the geometry transverse to the singular locus $\mathscr{C}_{\text{sing}}^{(r)}$ inside $\mathscr{C}$, where the intersection with the singular locus is the origin of the Coulomb branch of the $U(r)$ theory with $N_f$ flavours.

We conclude that the infrared physics at a generic point P in the singular locus $\mathscr{C}_{\text{sing}}^{(r)}$ corresponds to the interacting theory $T_{U(r),N_f}$, which is the infrared fixed point of a good theory, plus $N-r$ free twisted hypermultiplets,

$$P \in \mathscr{C}_{\text{sing}}^{(r)} \quad \xrightarrow{IR} \quad T_{U(r),N_f} + (N-r) \text{ free twisted hypers}. \tag{4.10}$$

This is the same as for good theories, the only difference being the range of $r$, which here is $0 \leq r \leq \lfloor \frac{N_f}{2} \rfloor$. In particular, near any point of the most singular locus $\mathscr{C}^*$, we find the $T_{U(\lfloor N_f/2 \rfloor),N_f}$ SCFT with $N - \lfloor \frac{N_f}{2} \rfloor$ free twisted hypermultiplets.

## 4.3 The full moduli space

As for good theories, the full moduli space is obtained by gluing together the Higgs and Coulomb branches. This is done by identifying the flat CB directions that open up at the subvariety $\mathscr{H}_r$ of the Higgs branch with the CB singular locus $\mathscr{C}_{\text{sing}}^{(r)}$, or equivalently by identifying the flat HB directions that open up at the subvariety $\mathscr{C}_{\text{sing}}^{(r)}$ of the Coulomb branch with $\mathscr{H}_r$. Indeed, the Higgs factor $\mathscr{H}_r$ is isomorphic to the Higgs branch $\mathscr{H}_{U(r),N_f}$ of the $U(r)$ theory with $N_f$ flavours. At a generic point in $\mathscr{H}_r$ the transverse Coulomb factor $\mathscr{C}_{N-r}$ in (4.1) is isomorphic to the Coulomb branch of a $U(N-r)$ theory with $N_f - 2r$ flavours, agreeing with $\mathscr{C}_{\text{sing}}^{(r)}$. Moreover, at the root (or origin) of $\mathscr{H}_r$, where there is enhanced $U(r)$ gauge symmetry, we should see that a Coulomb branch $\mathscr{C}_{U(r),N_f}$ opens up: this matches precisely the geometry transverse to $\mathscr{C}_{\text{sing}}^{(r)}$ inside $\mathscr{C}$ (4.9). This leads to the full moduli space

$$\mathscr{M} = \bigcup_{r=0}^{\lfloor \frac{N_f}{2} \rfloor} (\mathscr{C}_{N-r} \times \mathscr{H}_r), \tag{4.11}$$

with $\mathscr{C}_{N-r} = \mathscr{C}_{\text{sing}}^{(r)}$.

Near a generic point in $\mathscr{C}_{N-r} \times \mathscr{H}_r$ the theory is free, with $N-r$ free twisted hypermultiplets and $r(N_f - r)$ free ordinary hypermultiplets. At the root of $\mathscr{H}_r$ (but at a generic location in $\mathscr{C}_{N-r}$), the local geometry of the full moduli space takes the same form as in (2.38). This confirms that the infrared physics is described by (4.10), with a $T_{U(r),N_f}$ interacting SCFT and $N-r$ free twisted hypermultiplets.

## 4.4 Masses, FI parameters and moduli space of vacua

Like good theories, bad $U(N)$ SQCD with $N_f$ flavours of fundamental hypermultiplets have relevant deformation parameters: $N_f$ $SU(2)_C$ triplets of mass parameters, which are scalars for the background vector multiplets associated to the $SU(N_f)$ flavour symmetry, and one $SU(2)_H$ triplet of FI parameters, which are scalars for the background twisted vector multiplets associated to the $U(1)_J$ topological symmetry. As usual, we will focus on the complex parameters

that the complex algebraic geometry of the moduli space of vacua is sensitive to: the complex mass parameters $m_\alpha$, $\alpha = 1, \ldots, N_f$, and the complex FI parameter $\zeta$.

The effect of complex masses on the moduli space of vacua of bad theories can be analysed in complete analogy with the case of good theories presented in Section 2.5. The mass deformation deforms the Coulomb branch, and lifts the Higgs branch and all mixed branches, leaving only $SU(N_f)$-symmetric vacua in which the VEV of hypermultiplet scalars vanish.

The effect of the Fayet-Iliopoulos deformation presents some differences with the case of good theories, as we now explain in more detail. Let us start with the Coulomb branch. The FI deformation gives mass to the dynamical vector multiplet and lifts the flat directions of the Coulomb branch. The remaining supersymmetric Coulomb vacua are precisely the vacua invariant under the $U(1)_J$ topological symmetry, which are characterized by zero expectation values of monopole operators: $U^\pm(z) = 0$. Substituting in the Coulomb branch relations (2.17) at zero masses, we deduce that $Q(z) = z^N$ and $\tilde{Q}(z) = z^{N_f - N}$. This singles out a unique $U(1)_J$ invariant vacuum $\mathscr{P}$, in which the gauge invariant Coulomb branch operators take the following expectation values:

$$\underline{\mathscr{P}}: \quad \Phi_n = 0, \; 1 \le n \le N, \quad V_n^\pm = 0, \; 0 \le n \le N-1,$$
$$\widetilde{\Phi}_n = 0, \; n \neq 2N - N_f - 2, \quad \widetilde{\Phi}_{2N - N_f - 2} = (-1)^{N_f}. \tag{4.12}$$

The vacuum $\mathscr{P}$ is the bad theory analogue of the vacuum $\mathscr{C}^*$ that survives the FI deformation for good theories: it is the only Coulomb vacuum that preserves $U(1)_J$. While $\mathscr{C}^*$ is the origin of the Coulomb branch $\mathscr{C}$ of the good theory, which is algebraically a cone, we stress that the vacuum $\mathscr{P}$ is in no way the origin of the Coulomb branch of the bad theory, which algebraically is not a cone.[24]

Note that the $U(1)_J$ invariant vacuum $\mathscr{P}$ on the Coulomb branch only exists if $N_f \ge N$, because $\widetilde{Q}(z)$ is by definition a polynomial of $z$. For $N_f < N$, the topological symmetry is spontaneously broken everywhere on the Coulomb branch, and the FI deformation lifts the Coulomb branch entirely.

Next, we discuss the effect of the FI deformation on the Higgs branch. In Section 2.5 we saw that for good theories the FI parameter deformed the full Higgs branch (2.33) to a smooth manifold of the same quaternionic dimension $N(N_f - N)$, and lifted all the singular subvarieties of the Higgs branch which were part of mixed branches at zero FI parameter.

In the case of bad theories, instead, the Higgs branch of the undeformed theory has a top-dimensional component $\mathscr{H}_{\lfloor N_f/2 \rfloor}$ of quaternionic dimension $\lfloor N_f/2 \rfloor (N_f - \lfloor N_f/2 \rfloor)$ and is part of a mixed branch. This contains lower dimensional singular components $\mathscr{H}_r$ of dimension $r(N_f - r)$ with decreasing $r < \lfloor N_f/2 \rfloor$, which are part of mixed branches of increasing Coulomb branch dimension $N - r$. Turning on an FI parameter lifts most of these Higgs components, leaving only a smooth Higgs branch of dimension $(N_f - N)N$, which is a deformation of $\mathscr{H}_{N_f - N}$ and only exists if $N_f \ge N$.[25] If $N_f < N$, the FI deformation completely lifts the Higgs branch. We will focus on bad theories with $N_f \ge N$ in the following. On their deformed Higgs branch, the hypermultiplet scalars can be brought up to flavour and gauge transformations to the

---

[24]Like $\mathscr{C}^*$ for good theories, $\mathscr{P}$ is still a fixed point of the $\mathbb{C}^*$ action that complexifies $U(1)_R$, but the Coulomb branch is not a cone because not all its generators have positive $R$-charge.

[25]The surviving deformed component is again a hyperkähler quotient of the baryonic branch of the $SU(N)$ SQCD theory by the baryonic $U(1)_B$ symmetry at complex moment map $\zeta$. (The baryonic branch only exists for $N_f \ge N$.) The lifted components descend from non-baryonic branches.

form [18]

$$Q = \begin{pmatrix} 0 & \kappa_1 & 0 & 0 & \dots & 0 & 0 & 0 & \dots & 0 \\ 0 & 0 & 0 & \kappa_2 & \dots & 0 & 0 & 0 & \dots & 0 \\ \vdots & & & & \ddots & & & & & \vdots \\ 0 & 0 & 0 & 0 & \dots & 0 & \kappa_{\tilde{N}} & 0 & \dots & 0 \\ 0 & 0 & 0 & 0 & \dots & 0 & 0 & \kappa_0 & \dots & 0 \\ 0 & 0 & 0 & 0 & \dots & 0 & 0 & 0 & \ddots & 0 \\ 0 & 0 & 0 & 0 & \dots & 0 & 0 & 0 & \dots & \kappa_0 \end{pmatrix},$$

$$\tilde{Q}^T = \begin{pmatrix} \lambda_1 & \tilde{\kappa}_1 & 0 & 0 & \dots & 0 & 0 & 0 & \dots & 0 \\ 0 & 0 & \lambda_2 & \tilde{\kappa}_2 & \dots & 0 & 0 & 0 & \dots & 0 \\ \vdots & & & & \ddots & & & & & \vdots \\ 0 & 0 & 0 & 0 & \dots & \lambda_{\tilde{N}} & \tilde{\kappa}_{\tilde{N}} & 0 & \dots & 0 \\ 0 & 0 & 0 & 0 & \dots & 0 & 0 & \tilde{\kappa}_0 & \dots & 0 \\ 0 & 0 & 0 & 0 & \dots & 0 & 0 & 0 & \ddots & 0 \\ 0 & 0 & 0 & 0 & \dots & 0 & 0 & 0 & \dots & \tilde{\kappa}_0 \end{pmatrix},$$

(4.13)

with $\kappa_a \tilde{\kappa}_a = \zeta$ for all $a = 0, \cdots, \tilde{N} \equiv N_f - N$, ensuring the $F$-term equations $Q\tilde{Q} = \zeta \mathbb{1}_N$. The meson $M = \tilde{Q}Q$ satisfies $M^2 = \zeta M$ and takes the Jordan normal form

$$M = \begin{pmatrix} 0 & \lambda_1 \kappa_1 & 0 & 0 & \dots & 0 & 0 & 0 & \dots & 0 \\ 0 & \zeta & 0 & 0 & \dots & 0 & 0 & 0 & \dots & 0 \\ 0 & 0 & 0 & \lambda_2 \kappa_2 & \dots & 0 & 0 & 0 & \dots & 0 \\ 0 & 0 & 0 & \zeta & \dots & 0 & 0 & 0 & \dots & 0 \\ \vdots & & & & \ddots & & & & & \vdots \\ 0 & 0 & 0 & 0 & \dots & 0 & \lambda_{\tilde{N}} \kappa_{\tilde{N}} & 0 & \dots & 0 \\ 0 & 0 & 0 & 0 & \dots & 0 & \zeta & 0 & \dots & 0 \\ 0 & 0 & 0 & 0 & \dots & 0 & 0 & \zeta & \dots & 0 \\ \vdots & & & & & & & & \ddots & \vdots \\ 0 & 0 & 0 & 0 & \dots & 0 & 0 & 0 & \dots & \zeta \end{pmatrix},$$

(4.14)

with $N_f - N$ zero eigenvalues and $(N_f - N) + (2N - N_f) = N$ eigenvalues equal to $\zeta$.

It might come as a surprise that a lower-dimensional component $\mathcal{H}_{N_f - N}$ of the Higgs branch survives the FI deformation, while the top-dimensional component is lifted, together with all the other components. Recall however that all Higgs branches of the undeformed theory are part of mixed branches, which are generically lifted since the only Coulomb vacuum that survives the FI deformation is the $U(1)_J$ invariant vacuum $\mathscr{P}$. As we will emphasize in the next section, the point $\mathscr{P}$ belongs to the singular locus $\mathscr{C}_{2N - N_f}$ of the Coulomb branch, out of which a Higgs factor $\mathcal{H}_{N_f - N}$ of the mixed branch $\mathscr{C}_{2N - N_f} \times \mathcal{H}_{N_f - N}$ emanates. When the FI parameter $\zeta$ is turned on, all that remains of the full moduli space is the point $\mathscr{P}$ on the Coulomb branch, and the Higgs factor $\mathcal{H}_{N_f - N}$ that emanated from $\mathscr{P}$ becomes the smooth Higgs branch described above. If $N_f < N$, the FI deformation lifts the moduli space entirely, and supersymmetry is spontaneously broken.

If the FI parameter $\zeta$ is non-vanishing and the flavour masses $m_\alpha$ are all different, both the Coulomb and the Higgs branch of the moduli space of vacua are lifted, leaving at most isolated vacua which are fixed points of the action of the unbroken $U(1)_J \times U(1)^{N_f - 1}$ global symmetry. Here $U(1)_J$ is the topological symmetry that acts on the Coulomb branch, and $U(1)^{N_f - 1}$ is the maximal torus of the flavour symmetry $SU(N_f)$ which and acts on the Higgs branch of the theory with massless hypermultiplets.

To find the locations of these vacua on the Coulomb branch, we look for fixed points of the $U(1)_J$ action on the Coulomb branch deformed by the masses for the hypermultiplets. Monopole operators, which are charged under $U(1)_J$, must have zero VEV, and the generating polynomial of Coulomb branch relations reduces to

$$Q(z)\widetilde{Q}(z) = P(z) \equiv \prod_{\alpha=1}^{N_f}(z - m_\alpha) \,. \tag{4.15}$$

This equation requires $\widetilde{Q}(z)$ to be a monic polynomial of degree $N_f - N \geq 0$. There are $\binom{N_f}{N}$ isolated Coulomb vacua specified by which $N$ of the $N_f$ masses $m_\alpha$ are equal to the roots $\varphi_a$ of $Q(z)$, or equivalently which $N_f - N$ masses are equal to the roots of $\widetilde{Q}(z)$. The massless hypermultiplets $(Q^a{}_\alpha, (\widetilde{Q}^\dagger)^a{}_\alpha)$ which take expectation value in each of these vacua are those for which $\sigma_a = m_\alpha$. The meson matrix $M$ has a diagonal VEV with $N$ eigenvalues equal to $\zeta$ and the remaining $N_f - N$ eigenvalues equal to zero. When the masses go to zero, the $\binom{N_f}{N}$ isolated Coulomb vacua collapse to the symmetric vacuum $\mathscr{P}$, and a continuous Higgs branch opens up.

Note that for good or ugly theories, $\widetilde{Q}(z)$ is already a monic polynomial of degree $N_f - N$. For bad theories with $N \leq N_f \leq 2N - 2$, instead, $\widetilde{Q}(z)$ has degree $N - 2$, and the requirement that it reduces to a monic polynomial of lower degree $N_f - N$ sets $\widetilde{\Phi}_n = 0$ for $0 \leq n \leq 2N - N_f - 2$ and $\widetilde{\Phi}_{2N-N_f-2} = (-1)^{N_f}$. For $N_f < N$, the Coulomb branch relations (4.15) cannot be solved for all values of $z$ and supersymmetry is spontaneously broken, as we already observed for zero masses.

# 5   Seiberg non-duality

It was proposed in [8] that bad $U(N)$ SQCD theories with $N \leq N_f \leq 2N - 2$ flavours are infrared dual to the good $U(N_f - N)$ SQCD theories with $N_f$ flavours plus $2N - N_f$ free twisted hypermultiplets, realizing a Seiberg-like duality for $\mathcal{N} = 4$ theories. This claim is supported by the computations of the exact 3-sphere partition function [8], of the 2d twisted superpotential of the mass deformed $\mathcal{N} = 2^*$ theory on $\mathbb{R}^2 \times S^1$ [24], the supersymmetric index of the theory on $S^2$ (or $S^1 \times S^2$ partition function) [25], and recently of vortex partition functions [26] in the proposed dual theories. All these computations, however, apply to the theory deformed by an FI term. The precise claim remains obscure since it does not explain at which point(s) on the space of vacua of the undeformed flat space theory this infrared duality is supposed to occur. Even more puzzling is the observation that at zero FI parameter the Higgs branches of the would-be dual theories do not agree.[26] We will use our results on the infrared physics of bad theories to revisit the claims about Seiberg-like dualities. We will show that there is no exact duality, since the moduli space of vacua of the putative dual theories are different globally (we will find however that the moduli space of the good theory is embedded into the moduli space of the bad theory). Instead, what has been proposed as the dual of the bad theory is only the low energy effective description at the particular point $\mathscr{P}$ of the moduli space of vacua of the bad theory, in the spirit of [18]. We will also explain the results on partition functions found in the literature, which arise after one turns on an FI parameter. Finally we will identify the relation between the UV and IR R-symmetries in the special vacuum $\mathscr{P}$.

---

[26]But they do at non-zero FI parameter, where the Higgs branch is the cotangent bundle of the Grassmannian of $N$-planes (or $(N_f - N)$-planes) in $N_f$ dimensions: $Gr(N, N_f) \cong Gr(N_f - N, N_f)$.

## 5.1 Infrared effective theories and the symmetric vacuum

The structure of the moduli space of the bad SQCD theories studied in the previous section leads to a number of infrared effective theories that depend on the Coulomb vacuum.[27] The SQCD theory in a generic vacuum of the codimension $r$ singular locus $\mathscr{C}_{N-r} \equiv \mathscr{C}_{\text{sing}}^{(r)}$ flows to the low energy theory $T_{U(r),N_f}$ plus $N-r$ additional free twisted hypermultiplets. In particular the effective theory at the most singular locus $\mathscr{C}^* = \mathscr{C}_{N-\lfloor N_f/2 \rfloor}$ is $T_{U(\lfloor N_f/2 \rfloor),N_f}$ with $N - \lfloor \frac{N_f}{2} \rfloor$ free twisted hypermultiplets.

From this perspective the "Seiberg dual" theory proposed in [8] is the same as the low energy effective theory at any generic point on the singular locus $\mathscr{C}_{2N-N_f}$ of the Coulomb branch $\mathscr{C}$, where the infrared CFT is $T_{U(N_f-N),N_f}$ with $2N-N_f$ free twisted hypermultiplets. However there seems to be no particular reason at this point to distinguish this infrared duality from the others arising at different locations on $\mathscr{C}$.

Let us call $\mathscr{C}_b$ the Coulomb branch of the bad theory with gauge group $U(N)$ and $N_f$ flavours in the range $N \leq N_f \leq 2N-2$ and $\mathscr{C}_g$ the Coulomb branch of the the good theory with gauge group $U(N_f - N)$ and $N_f$ flavours. As before we set to zero all masses. An interesting observation is that the Coulomb branch $\mathscr{C}_g$ is a subvariety of the Coulomb branch $\mathscr{C}_b$:[28]

$$\mathscr{C}_g \subset \mathscr{C}_b \,. \tag{5.1}$$

This can be seen as follows. The space $\mathscr{C}_b$ is described by

$$Q_N(z)\widetilde{Q}_{N-2}(z) + U_{N-1}^+(z)U_{N-1}^-(z) = z^{N_f} \,, \tag{5.2}$$

where the indices indicate the degree in $z$ of the polynomials and $Q_N(z)$ is a monic polynomial. The space $\mathscr{C}_g$ on the other hand is described by the polynomial relation

$$\mathbf{Q}_{N_f-N}(z)\widetilde{\mathbf{Q}}_N(z) + \mathbf{U}_{N_f-N-1}^+(z)\mathbf{U}_{N_f-N-1}^-(z) = z^{N_f} \,, \tag{5.3}$$

with $\mathbf{Q}_{N_f-N}(z)$ a monic polynomial. This relation implies that $\widetilde{\mathbf{Q}}_N(z)$ is monic too. The embedding (5.1) is found by solving (5.2) with[29]

$$Q_N(z) = \widetilde{\mathbf{Q}}_N(z), \quad \widetilde{Q}_{N-2}(z) = \mathbf{Q}_{N_f-N}(z), \quad U_{N-1}^\pm(z) = \mathbf{U}_{N_f-N-1}^\mp(z). \tag{5.4}$$

The polynomial relation (5.3) then implies that the polynomial relation (5.2) is satisfied. The subvariety (isomorphic to) $\mathscr{C}_g$ inside $\mathscr{C}_b$ is described as

$$
\begin{aligned}
\mathscr{C}_g \cong \mathscr{C}_b \cap \{ \, & V_n^\pm = 0, \quad 0 \leq n \leq 2N-N_f-1, \\
& \widetilde{\Phi}_n = 0, \quad 0 \leq n \leq 2N-N_f-3, \\
& \widetilde{\Phi}_{2N-N_f-2} = (-1)^{N_f} \, \}.
\end{aligned} \tag{5.5}
$$

Comparing with the description of the singular loci $\mathscr{C}_{b,N-r} \equiv \mathscr{C}_{b,\text{sing}}^{(r)}$ of the bad theory, we deduce that the subvariety $\mathscr{C}_g \subset \mathscr{C}_b$ intersects all the singular loci $\mathscr{C}_{b,N-r}$ with $1 \leq r \leq N_f-N$, but does not intersect the singular loci of codimension $r > N_f - N$:

$$
\mathscr{C}_g \cap \mathscr{C}_{b,N-r} \begin{cases} \neq \emptyset & \text{for} \quad 0 \leq r \leq N_f - N, \\ = \emptyset & \text{for} \quad N_f - N + 1 \leq r. \end{cases} \tag{5.6}
$$

---

[27]We will always sit at the origin $\mathscr{H}^*$ of the Higgs branch, where all hypermultiplet scalars vanish.

[28]This was pointed out to us by D. Gaiotto.

[29]At the level of the Coulomb branch analysis, the mapping of topological charges has a sign ambiguity. Our choice here anticipates the analysis of the sphere partition function in Section 5.2.

On the subvariety $\mathscr{C}_g$ there is a distinguished point $\mathscr{P} \equiv \mathscr{C}_g^*$ which is the origin of $\mathscr{M}_g$, where the $T_{U(N_f-N),N_f}$ SCFT lives. This is nothing but the $U(1)_J$ invariant Coulomb vacuum discussed in Section 4.4. This point in $\mathscr{C}_g$ is defined by $\mathbf{Q}_{N_f-N}(z) = z^{N_f-N}$, $\widetilde{\mathbf{Q}}_N(z) = z^N$ and $\mathbf{U}_{N_f-N-1}^+(z) = 0$. In the Coulomb branch $\mathscr{C}_b$ the point $\mathscr{P}$ is described by

$$\underline{\mathscr{P}} : \quad Q_N(z) = z^N, \quad \widetilde{Q}_{N-2}(z) = z^{N_f-N}, \quad U_{N-1}^\pm(z) = 0. \tag{5.7}$$

It is easy to see that $\mathscr{P}$ is the point of intersection of the subvariety $\mathscr{C}_g$ and the codimension $N_f - N$ singular locus $\mathscr{C}_{b,2N-N_f}$ inside $\mathscr{C}_b$:

$$\mathscr{P} = \mathscr{C}_g \cap \mathscr{C}_{b,2N-N_f}. \tag{5.8}$$

Hence the geometry of $\mathscr{C}_b$ close to $\mathscr{P}$ is

$$\mathscr{U}[\mathscr{P}] = \mathscr{C}_{U(N_f-N),N_f} \times \mathbb{C}^{2(2N-N_f)} \equiv \mathscr{C}_g \times \mathbb{C}^{2(2N-N_f)}. \tag{5.9}$$

The geometry transverse to $\mathscr{C}_g$ at the point $\mathscr{P}$ is that of $2N-N_f$ free twisted hypermultiplets.

A similar conclusion holds for Higgs branches. The classical analysis reviewed in Sections 2.4 and 4 shows that the Higgs branch $\mathscr{H}_g$ of the good $U(N_f - N)$ theory with $N_f$ flavours is embedded in the Higgs branch $\mathscr{H}_b$ of the bad $U(N)$ theory with $N_f$ flavours:

$$\mathscr{H}_g \cong \overline{\mathscr{O}}_{(2^{N_f-N},1^{2N-N_f})} \subset \mathscr{H}_b \cong \overline{\mathscr{O}}_{(2^{\lfloor N_f/2 \rfloor},1^{N_f \bmod 2})}. \tag{5.10}$$

Taking mixed branches also into account, one can see that the full moduli space $\mathscr{M}_g$ of the good theory is embedded in the full moduli space $\mathscr{M}_b$ of the bad theory. The local geometry of $\mathscr{M}_b$ near $\mathscr{P}$ (which corresponds to the origin of the full moduli space $\mathscr{M}_g$ of the good theory) takes the form

$$\mathscr{U}_{tot}[\mathscr{P}] = \mathscr{M}_g \times \mathbb{C}^{2(2N-N_f)}, \tag{5.11}$$

reproducing the full moduli space of $T_{U(N_f-N),N_f}$ plus $2N-N_f$ free twisted hypermultiplets. This shows that the "Seiberg dual" theory of [8] is in fact the low energy effective description at the point $\mathscr{P}$: the bad $U(N)$ SQCD theory with $N_f$ flavours in the distinguished vacuum $\mathscr{P} \in \mathscr{C}_b$ flows to the $T_{U(N_f-N),N_f}$ infrared CFT with $2N-N_f$ free twisted hypermultiplets,

$$\text{vacuum } \mathscr{P} \quad \xrightarrow{IR} \quad T_{U(N_f-N),N_f} + (2N-N_f) \text{ free twisted hypers}. \tag{5.12}$$

Notice that the point $\mathscr{P}$ does not belong to the most singular locus of the bad theory $T_b$, but rather it is a generic point in $\mathscr{C}_{2N-N_f}$. It is perhaps a surprise of our analysis of bad theories that the point where all $\Phi_n$ and $V_n^\pm$ vanish is *not* the most singular point in the Coulomb branch geometry (unlike for good theories).

We remark that the vacuum $\mathscr{P}$ preserves all the global symmetries of the theory, and not only $U(1)_J$. We will therefore refer to $\mathscr{P}$ as the *symmetric vacuum*.

## 5.2 FI parameter, $S^3$ partition function, and $R$-symmetry

We have found that the proposed Seiberg duality does not hold globally on the space of vacua of the bad theory, but it is a local effective description in the vicinity of the symmetric vacuum $\mathscr{P}$. We will explain in this section the results on sphere partition functions found in [8], which arise when a non-zero FI parameter is turned on, but first we need to identify the chiral operators decoupling in the infrared theory at $\mathscr{P}$.

The free twisted hypermultiplets in the infrared theory at the symmetric vacuum $\mathscr{P}$ arise from the fluctuations of operators which parametrize the space tangent to $\mathscr{C}_{b,2N-N_f} \subset \mathscr{C}_b$ at

$\mathscr{P}$. Let us denote the operator fluctuations $\delta\Phi_{n=1,\cdots,N}, \delta\widetilde{\Phi}_{n=0,\cdots,N-2}, \delta V^\pm_{n=0,\cdots,N-1}$. The CB relations (4.4) of the bad theory reduce as follows when they are expanded near the vacuum $\mathscr{P}$. The first $2N-N_f-1$ relations serve to fix the fluctuations $\delta\widetilde{\Phi}_{n=0,\cdots,2N-N_f-2}$, the next $N$ relations serve to fix $\delta\Phi_{n=1,\cdots,N}$, using $\widetilde{\Phi}_{2N-N_f-2} \sim 1$. The remaining $N_f - N$ relations are identified with the CB relations of the good $U(N_f - N)$ theory with $N_f$ flavours, which arise after solving for the $\widetilde{\Phi}$ operators in the good theory. The fluctuations identified with the CB operators of the good theory are $\delta\widetilde{\Phi}_{n=2N-N_f-1,\cdots,N-2}$ and $\delta V^\pm_{2N-N_f,\cdots,N-2}$, corresponding to the $\Phi$ and $V^\mp$ operators of the good theory respectively. A detailed analysis for the $U(3)$ theory with $N_f = 4$ flavours is presented in appendix A.2.

The fluctuations of the monopole operators $\delta V^\pm_{n=0,\cdots,2N-N_f-1}$ decouple from the equations and parametrize the $2N - N_f$ free twisted hypermultiplets of the infrared theory. From the $R$-charge formula

$$R[V^\pm_n] = \frac{N_f}{2} - N + 1 + n, \qquad (5.13)$$

we see that the monopole operators which decouple contain all the operators of negative or vanishing $R$-charge $\delta V^\pm_{n=0,\cdots,N-\lfloor N_f/2\rfloor-1}$, as well as some operators of positive $R$-charge.

Turning on a non-zero FI parameter $\zeta$ lifts the Coulomb branch to the symmetric vacuum $\mathscr{P}$ and gives a complex mass $\zeta$ to all the free twisted hypermultiplets, which have charge one under the topological $U(1)_J$ that is weakly gauged. The effective theory at the only surviving Coulomb vacuum $\mathscr{P}$, when $\zeta \neq 0$, is then the $T_{U(N_f-N),N_f}$ fixed point deformed by the FI term with parameter $-\zeta$ along with $2N - N_f$ free massive twisted hypermultiplets with mass $\zeta$.

This explains the observations of [8] about sphere partition functions. It was observed that the exact sphere partition function of the bad theory $Z_{N,N_f}$, defined by choosing a suitable choice of integration contour of the matrix integral,[30] and the sphere partition function of the good theory $Z_{N_f-N,N_f}$ satisfy an identity[31] which at zero flavour masses can be recast into the form

$$Z_{N,N_f}(\eta) = Z_{N_f-N,N_f}(-\eta) \prod_{n=0}^{2N-N_f-1} Z_{\text{chiral}}(\eta, r_n) Z_{\text{chiral}}(-\eta, r_n) \qquad (\eta \neq 0),$$

$$\text{with} \quad r_n = \frac{N_f}{2} - N + 1 + n, \qquad (5.14)$$

where $\eta$ is the real FI parameter in the bad theory, $-\eta$ is the real FI parameter of the good theory, and $Z_{\text{chiral}}(m, r)$ is the sphere partition function of a free chiral multiplet of real mass $m$ and $R$-charge $r$. This is in complete agreement with our results: on the sphere the matter fields acquire a mass due to the coupling to the background gravity multiplet, lifting the deformed Higgs branch, therefore the theory on the sphere at non-zero FI parameter has a single vacuum, identified with the symmetric vacuum $\mathscr{P}$ (the root of the deformed Higgs branch) in the limit of large sphere radius. The partition function is independent of the sphere radius, thus we expect from our analysis that the partition function of the bad theory will match the partition function of the good theory multiplied by the partition function of $2N-N_f$ free twisted hypermultiplets, which arise from (a rearrangement of) the $2N - N_f$ couples of chiral multiplets $(\delta V^+_n, \delta V^-_n)$ with opposite masses $\eta, -\eta$ and identical $R$-charges $r_n$, $n = 0, \cdots, 2N-N_f-1$. This is precisely

---

[30]For good and ugly theories the matrix model computing the sphere partition function is convergent for any real value of $\eta$. For bad theories the matrix model integral on the physical contour is divergent, however, for $\eta \neq 0$, it can be regularized by changing integration contour. It is this regularized matrix integral which satisfies the identity (5.14). This is explained in [8] (see footnote 2), summarizing results derived in [9].

[31]We spotted a typo in formula (3.35) in [8] which should read $\eta_u = -\eta - (2N_c - N_f - 1)\frac{\omega}{2}$.

the content of (5.14).[32]

The $R$-charges appearing in the above expressions are those under the $U(1)$ $R$-symmetry that is manifest in the UV and is used to define the sphere partition function of the bad theory. As emphasized in [8], this differs from the infrared superconformal $R$-charge, which should be one-half for the chiral multiplets in free hypermultiplets, which saturate the unitarity bound. Let us denote $U(1)_{\text{UV}}$ and $U(1)_{\text{IR}}$ the UV and IR $R$-symmetries. In the symmetric vacuum $\mathcal{P}$ all operators charged under $U(1)_{\text{UV}}$ are set to zero, therefore $U(1)_{\text{UV}}$ is not spontaneously broken and is a symmetry of the infrared theory as well. In addition there is an accidental infrared $U(2N - N_f)_K$ global symmetry rotating the $(2N - N_f)$ free hypermultiplets with equal mass $\eta$ (and $\zeta$). Under $U(2N - N_f)_K$, the chiral multiplets of mass $\eta$ transform in the fundamental representation and the chiral multiplets of masses $-\eta$ transform in the anti-fundamental representation. Moreover all chiral multiplets must have canonical super-conformal $R$-charge one-half under $U(1)_{\text{IR}}$.

It is then relatively straightforward to identify the relation between $U(1)_{\text{UV}}$ and $U(1)_{\text{IR}}$ in the infrared theory, in order to reproduce the $U(1)_{\text{UV}}$ $R$-charges assigned to the decoupling chiral multiplets in equation (5.14). We find that

$$U(1)_{\text{UV}} = \text{diag}(U(1)_{\text{IR}} \times U(1)_K), \tag{5.15}$$

where $U(1)_K$ denotes the $U(1) \hookrightarrow U(2N - N_f)_K$ embedding under which the fundamental representation decompose into $(N - \frac{N_f}{2} - \frac{1}{2}, N - \frac{N_f}{2} - \frac{3}{2}, \cdots, -N + \frac{N_f}{2} + \frac{1}{2})$. The $U(1)_{\text{UV}}$ $R$-charges of a pair of chiral multiplets forming a free twisted hypermultiplet of the infrared theory are then

$$r^+_{\text{UV},n} = N - \frac{N_f}{2} - n, \quad r^-_{\text{UV},n} = \frac{N_f}{2} - N + 1 + n, \quad n = 0, \cdots, 2N - N_f - 1. \tag{5.16}$$

This matches the $R$-charges in (5.14), which can be rewritten as

$$Z_{N,N_f}(\eta) = Z_{N_f - N, N_f}(-\eta) \prod_{n=0}^{2N - N_f - 1} Z^{(n)}_{\text{hyper}}(\eta) \qquad (\eta \neq 0), \tag{5.17}$$

$$Z^{(n)}_{\text{hyper}}(\eta) = Z_{\text{chiral}}(-\eta, r_n) Z_{\text{chiral}}(\eta, r_{2N - N_f - 1 - n}) = Z_{\text{chiral}}(-\eta, r^-_{\text{UV},n}) Z_{\text{chiral}}(\eta, r^+_{\text{UV},n}).$$

This is not the whole story since with $\mathcal{N} = 4$ supersymmetry the chiral multiplets in a twisted hypermultiplet are doublets of an $SU(2)_{\text{IR}}$ $R$-symmetry, whose Cartan generator is $U(1)_{\text{IR}}$. The symmetric vacuum $\mathcal{P}$ manifestly preserves $U(1)_{\text{UV}}$ and, since the choice of $U(1)_{\text{UV}} \subset SU(2)_{\text{UV}} \equiv SU(2)_C$ is arbitrary, it must be that the full $SU(2)_{\text{UV}}$ is unbroken. Thus the infrared theory preserves $SU(2)_{\text{UV}}$. We conclude that $SU(2)_{\text{UV}}$ must be a combination of $SU(2)_{\text{IR}}$ and an $SU(2)_K \subset U(2N - N_f)_K$ accidental symmetry in the infrared theory. The above considerations lead to the identification

$$SU(2)_{\text{UV}} = \text{diag}(SU(2)_{\text{IR}} \times SU(2)_K), \tag{5.18}$$

with $SU(2)_K$ the principal embedding of $SU(2)$ inside $U(2N - N_f)_K$, namely the embedding associated to the partition $[2N - N_f]$ of $2N - N_f$.[33]

---

[32]The FI parameter appearing in the sphere partition function is actually a real FI parameter, whereas we have been studied the deformation of the space of vacua due to complex FI terms. This does not affect the discussion. The FI parameters form a triplet under the $\mathcal{N} = 4$ $SU(2)_H$ $R$-symmetry and the qualitative results are independent of which component of the triplet is chosen.

[33]$U(1)_K$ is generated by $\frac{1}{2}\tau_3$ with $\tau_3 = \text{diag}(1,-1) \in \mathfrak{su}(2)_K$.

We conclude that the decoupling monopole operators combine into free twisted hypermultiplets ($\delta V_n^-, \delta V_{2N-N_f-1-n}^+$). An explicit example is detailed in Appendix A.2. This identification was already proposed in [25], based on the computation of the (regularized) supersymmetric index of the bad theory using its factorization into vortex partition functions and an identity relating it to the index of the putative dual (as can be done with the sphere partition function). Notice that, contrary to the naive expectation, the chiral monopole operators of negative or zero $U(1)_{\text{UV}}$ $R$-charges are not the only operators which decouple. In fact each such chiral monopole operator is paired with a monopole operator of positive $U(1)_{\text{UV}}$ $R$-charge (which does not violate the $\mathcal{N} = 2$ unitarity bound) to make a free twisted hypermultiplet in the infrared theory.[34]

In this section we have explained the relation between sphere partition functions found in [8]. Similarly, other exact results [24–26], computed at non-vanishing FI parameter, are explained by the observation that the Coulomb branch is lifted to the symmetric vacuum $\mathscr{P}$ in those cases.

# 6 Future directions

In this paper we have analysed the quantum moduli space of vacua of 3d $\mathcal{N} = 4$ $U(N)$ SQCD theories, in particular in the bad regime of parameters $N_f \leq 2N-2$. This allowed us to describe the low-energy effective theories at singular loci on the Coulomb branch as infrared fixed points of good theories plus free fields, and to revisit and correct the claims about Seiberg-like duality for $\mathcal{N} = 4$ theories. There are many interesting directions one can explore from this starting point and we will list only a few here.

First, this analysis can be repeated for 3d $\mathcal{N} = 4$ SQCD theories with classical gauge groups. To do so, one should further develop the method of [17] to determine the Coulomb branch geometry in terms of the VEV of gauge invariant operators. For orthogonal and symplectic groups we expect a simple description to exist, since the Coulomb branch is a complete intersection [15]. For $SU(N)$ gauge group, the Coulomb branch is not a complete intersection, but it can be obtained as a hyperkähler quotient of the $U(N)$ Coulomb branch by the topological $U(1)_J$ symmetry that acts on monopole operators.[35]

Secondly, it would be interesting to generalise our analysis to circular quivers with unitary gauge groups, which have holographic dual solutions exhibiting cascading RG flows and enhançons [27–29]. We expect that these cascading RG flows are explained by the physics of the symmetric vacuum in the dual 3d $\mathcal{N} = 4$ circular quivers, analogously to the rôle of the baryonic root in explaining the holographic RG flows of [30, 31] for 4d $\mathcal{N} = 2$ circular quivers [32, 33].

Another direction is to study the space of vacua of 3d $\mathcal{N} = 4$ theories of Chern-Simons type [34, 35], which should be of the same form as $\mathcal{N} = 4$ Yang-Mills theories, but with branches where both monopole operators and matter scalars take VEV (see e.g. [36, 37]). This could reveal new dualities between the infrared fixed points of Yang-Mills quivers and Chern-Simons SCFTs.

Finally, it was found in [18] that Seiberg duality of 4d $\mathcal{N} = 1$ SQCD (with a quartic superpotential) can be understood from the low-energy limit of $\mathcal{N} = 2$ SQCD softly broken to $\mathcal{N} = 1$ by a mass for the vector multiplet scalar. It would be interesting to study the analogous situation in three dimensions, breaking $\mathcal{N} = 4$ supersymmetry to $\mathcal{N} = 2$ by a complex or real mass, and see if this leads at low energies to Seiberg-like or level-rank-like dualities of 3d $\mathcal{N} = 2$ SQCD theories [38–40].

We hope to report our progress in some of these directions in the near future.

---

[34]The spectrum of UV $R$-charges of the chiral monopole operators which decouple is symmetric around one-half.

[35]Gauging the topological $U(1)_J$ symmetry is equivalent to partially freezing $U(N)$ to $SU(N)$.

## Acknowledgements

We thank Andreas Braun, Cyril Closset, Nick Dorey, Davide Gaiotto, Amihay Hanany, Paul Heslop and Ronen Plesser for fruitful discussions at various stages of the project.

## A   Geometry near singular submanifolds

In this appendix we explicitly analyse the geometry near singular loci $\mathscr{C}_{N-r} \equiv \mathscr{C}_{\text{sing}}^{(r)}$ in two examples, using only gauge invariant operators. The analysis will confirm the result (2.31). The first example is that of a good theory. The second example is a bad theory and we analyse the geometry near the singular locus whose low energy effective theory is the one proposed as a Seiberg-like dual theory in [8].

### A.1   $U(2)$ with $N_f = 4$

First we look at the $U(2)$ SQCD theory with $N_f = 4$ flavours. This is a good theory. The CB relations (2.21) (at zero complex masses) are

$$
\begin{aligned}
\widetilde{\Phi}_0 &= 1, \\
\widetilde{\Phi}_1 + \Phi_1 \widetilde{\Phi}_0 &= 0, \\
\widetilde{\Phi}_2 + \Phi_1 \widetilde{\Phi}_1 + \Phi_2 \widetilde{\Phi}_0 + V_0^+ V_0^- &= 0, \\
\Phi_1 \widetilde{\Phi}_2 + \Phi_2 \widetilde{\Phi}_1 + V_0^+ V_1^- + V_0^- V_1^+ &= 0, \\
\Phi_2 \widetilde{\Phi}_2 + V_1^+ V_1^- &= 0.
\end{aligned}
\tag{A.1}
$$

The geometry close to a generic point in the codimension one singular locus $\mathscr{C}_{\text{sing}}^{(1)}$ can be obtained by taking the operators $\Phi_2, \widetilde{\Phi}_2, V_1^\pm$ to be of order $\epsilon \ll 1$:[36]

$$
\Phi_2 = O(\epsilon), \quad \widetilde{\Phi}_2 = O(\epsilon), \quad V_1^\pm = O(\epsilon).
\tag{A.2}
$$

To be away from the more singular locus $\mathscr{C}_{\text{sing}}^{(2)}$, we need to assume that at least one operator among $\Phi_1, \widetilde{\Phi}_1$ and $V_0^\pm$ is of order $\epsilon^0$. After solving for $\widetilde{\Phi}_0$ and $\widetilde{\Phi}_1$, we can rewrite the relations as

$$
\begin{aligned}
-(\Phi_1)^2 + V_0^+ V_0^- &= -\widetilde{\Phi}_2 - \Phi_2 \\
\Phi_1 \widetilde{\Phi}_2 - \Phi_2 \Phi_1 + V_0^+ V_1^- + V_0^- V_1^+ &= 0, \\
\Phi_2 \widetilde{\Phi}_2 + V_1^+ V_1^- &= 0.
\end{aligned}
\tag{A.3}
$$

Let us choose a generic vacuum on the singular submanifold with $\Phi_1$ of order $\epsilon^0$, $\Phi_1 = O(\epsilon^0)$. We can then solve for $\widetilde{\Phi}_2$ using the equation in the second line $\widetilde{\Phi}_2 = \Phi_2 - (\Phi_1)^{-1}(V_0^+ V_1^- + V_0^- V_1^+)$. Using this expression and the relation in the first line, one can recast the third relation as

$$
\mathbf{u}^+ \mathbf{u}^- = \mathbf{\Phi}^4 (1 + O(\epsilon)),
\tag{A.4}
$$

with

$$
\mathbf{u}^\pm = V_1^\pm - \frac{\Phi_2}{\Phi_1} V_0^\pm - \frac{\Phi_2^2}{\Phi_1^3} V_0^\pm - 2 \frac{\Phi_2^3}{\Phi_1^5} V_0^\pm, \quad \mathbf{\Phi} = \frac{\Phi_2}{\Phi_1}.
\tag{A.5}
$$

---

[36]The scaling of the operators with $\epsilon$ corresponds to taking one triple $(\varphi_a, u_a^+, u_a^-)$ to be of order $\epsilon$.

In the limit $\epsilon \to 0$, this reproduces the CB relation of the $U(1)$ theory with $N_f = 4$ flavours, that we are probing at its origin. This leaves the first relation in (A.3), which can be written more suggestively as

$$\frac{V_0^+}{\Phi_1} \frac{V_0^-}{\Phi_1} = 1 + O(\epsilon), \quad \delta\Phi_1 \text{ free}, \tag{A.6}$$

where $\delta\Phi_1$ denotes the fluctuation about the VEV $\Phi_1$. In the limit $\epsilon \to 0$, this agrees with the smooth Coulomb branch $\mathbb{C} \times \mathbb{C}^*$ of the free $U(1)$ theory, which is locally isomorphic to $\mathbb{C}^2$ around any point. The geometry near a generic point in $\mathscr{C}_{\text{sing}}^{(1)}$ is then of the form

$$\mathscr{U}[\mathscr{C}_{\text{sing}}^{(1)}] = \mathscr{C}_{U(1),4} \times \mathbb{C}^2, \tag{A.7}$$

in agreement with (2.31).

The codimension two singular locus $\mathscr{C}_{\text{sing}}^{(2)}$ is the most singular locus of the $U(2)$ theory with $N_f = 4$ flavours and is a single point $\mathscr{C}^*$, as for all good theories, where the theory should flow to the interacting fixed point $T_{U(2),4}$ without decoupling hypermultiplets, according to (2.31). At the level of CB relations, this means that the geometry is invariant under the scaling symmetry about the point $\mathscr{C}^*$, so that the geometry near $\mathscr{C}^*$ is the same as $\mathscr{C}_{U(2),4}$. It is easy to see that indeed the CB relations are invariant under the rescaling

$$(\Phi_1, \widetilde{\Phi}_1, V_0^\pm) \to \epsilon(\Phi_1, \widetilde{\Phi}_1, V_0^\pm), \quad (\Phi_2, \widetilde{\Phi}_2, V_1^\pm) \to \epsilon^2(\Phi_1, \widetilde{\Phi}_1, V_0^\pm), \tag{A.8}$$

which corresponds to zooming in on the origin $\mathscr{C}^*$ of $\mathscr{C}_{U(2),4}$.

## A.2 $U(3)$ with $N_f = 4$

The second example is the $U(3)$ theory with $N_f = 4$ flavours. The CB relations are

$$\begin{aligned}
\widetilde{\Phi}_0 + V_0^+ V_0^- &= 1, \\
\widetilde{\Phi}_1 + \Phi_1 \widetilde{\Phi}_0 + V_0^+ V_1^- + V_0^- V_1^+ &= 0, \\
\Phi_2 \widetilde{\Phi}_0 + \Phi_1 \widetilde{\Phi}_1 + V_0^+ V_2^- + V_0^- V_2^+ + V_1^+ V_1^- &= 0, \\
\Phi_3 \widetilde{\Phi}_0 + \Phi_2 \widetilde{\Phi}_1 + V_1^+ V_2^- + V_1^- V_2^+ &= 0, \\
\Phi_3 \widetilde{\Phi}_1 + V_2^+ V_2^- &= 0.
\end{aligned} \tag{A.9}$$

We study the geometry close to the codimension one singular locus $\mathscr{C}_{\text{sing}}^{(1)}$, to which the symmetric vacuum discussed in Section 5 belongs. The geometry near a generic point is obtained by letting the operators $\Phi_3, \widetilde{\Phi}_1, V_2^\pm$ be of order $\epsilon$,

$$\Phi_3 = O(\epsilon), \quad \widetilde{\Phi}_1 = O(\epsilon), \quad V_2^\pm = O(\epsilon), \tag{A.10}$$

with $\epsilon \ll 1$, and keeping at least one of the operators $\Phi_2, \widetilde{\Phi}_0, V_1^\pm$ to be of order $\epsilon^0$.

The first three relations in (A.9), when $\epsilon$ goes to zero, describe the Coulomb branch of a $U(2)$ theory with two flavours, that we are probing around a generic point, so that the fluctuations of the operators involved in the equations describe two free twisted hypermultiplets parametrizing $\mathbb{C}^4$. This leaves us with the last two equations constraining the operators $\Phi_3, \widetilde{\Phi}_1$ and $V_2^\pm$, which are of order $\epsilon$.

Let us assume $\widetilde{\Phi}_0 = O(\epsilon^0)$. Then the second, third and fourth equations can be used to solve for $\Phi_1, \Phi_2$ and $\Phi_3$ respectively. Plugging this in the fifth equation, one finds after some manipulations

$$\mathbf{u}^+ \mathbf{u}^- = \mathbf{\Phi}^4, \quad \text{with} \quad \mathbf{u}^\pm = V_2^\pm - \frac{\widetilde{\Phi}_1}{\widetilde{\Phi}_0} V_1^\pm + \frac{\widetilde{\Phi}_1^2}{\widetilde{\Phi}_0^2} V_0^\pm, \quad \mathbf{\Phi} = \frac{\widetilde{\Phi}_1}{\widetilde{\Phi}_0}. \tag{A.11}$$

|  | $U(1)_{\text{IR}}$ | $U(1)_K$ | $U(1)_{\text{UV}}$ |
|---|---|---|---|
| $V_0^\pm$ | $\frac{1}{2}$ | $-\frac{1}{2}$ | $0$ |
| $V_1^\pm$ | $\frac{1}{2}$ | $\frac{1}{2}$ | $1$ |

Table 1

This is the Coulomb branch of the $U(1)$ theory with $N_f = 4$ flavours, that we probe at the scale invariant point, the origin. The local geometry is therefore $\mathscr{C}_{U(1),4} \times \mathbb{C}^4$ in agreement with (2.31).

The symmetric vacuum $\mathscr{P}$ (where all global symmetries are preserved) has also $\widetilde{\Phi}_0 = 1$, $V_1^\pm = 0$, $V_0^\pm = 0$ (see Section 5). The map of operators with those of the effective $T_{U(1),4}$ theory at this point is simply $\mathbf{u}^\pm = V_2^\pm$, $\boldsymbol{\Phi} = \widetilde{\Phi}_1$. The fluctuations of $\widetilde{\Phi}_0$, $\Phi_1$ and $\Phi_2$ are fixed by the first three CB equations, leaving the fluctuations of $V_0^\pm$ and $V_1^\pm$ free and parametrizing the $\mathbb{C}^4$ space tangent to $\mathscr{C}_{\text{sing}}^{(1)}$ at $\mathscr{P}$. This means that the (fluctuations of the) monopole operators $V_0^\pm$ and $V_1^\pm$ become the complex scalars of free twisted hypermultiplets. Their superconformal $U(1)_{\text{IR}}$ $R$-charges have to be $\frac{1}{2}$. This does not match their UV $U(1)_{\text{UV}}$ $R$-charge which is 0 for $V_0^\pm$ and 1 for $V_1^\pm$. We conclude that the $U(1)_{\text{IR}}$ $R$-charge is a combination of the $U(1)_{\text{UV}}$ $R$-symmetry, which is preserved at the point $\mathscr{P}$, and accidental global symmetries under which the monopoles are charged. The accidental global symmetry here is the $U(2)_K$ global symmetry which rotates the two free twisted hypermultiplets as a doublet. The $U(1)$ $R$-symmetry therefore mixes with a Cartan $U(1)_K \subset SU(2)_K$:

$$U(1)_{\text{UV}} = \text{diag}(U(1)_{\text{IR}} \times U(1)_K), \tag{A.12}$$

with the charges given in Table 1. Since the infrared fixed point has $\mathcal{N} = 4$ supersymmetry, $U(1)_{\text{IR}}$ is only a subgroup of the $SU(2)_{\text{IR}}$ super-conformal $R$-symmetry acting on the Coulomb branch and the monopole operators in twisted hypermultiplets organise into complex doublets of $SU(2)_{\text{IR}}$. We observe that $(V_1^+, V_0^{-\dagger})$ is a doublet of $SU(2)_{\text{IR}}$ with $U(1)_K$ charge $+\frac{1}{2}$ and $(V_1^{-\dagger}, V_0^+)$ is another doublet of $SU(2)_{\text{IR}}$ with $U(1)_K$ charge $-\frac{1}{2}$. They make a free twisted hypermultiplet transforming in the representation $\mathbf{2_1}$ of the global symmetry $U(2)_K$.

In addition, the full $SU(2)_{\text{UV}}$ $R$-symmetry is preserved along the RG flow at $\mathscr{P}$ and is distinct from $SU(2)_{\text{IR}}$. Hence the $SU(2)_{\text{UV}}$ symmetry must be a combination of $SU(2)_{\text{IR}}$ and the accidental $SU(2)_K \subset U(2)_K$ global symmetry which arise at the infrared fixed point. This leads to

$$SU(2)_{\text{UV}} = \text{diag}(SU(2)_{\text{IR}} \times SU(2)_K). \tag{A.13}$$

Under $SU(2)_{\text{UV}}$ the fields decompose into the complex representations $\mathbf{3} + \mathbf{1}$, with the triplet $(V_1^+, V_0^+ + V_0^{-\dagger}, V_1^{-\dagger})$ and the singlet $V_0^+ - V_0^{-\dagger}$. It is not clear in which multiplet of the UV $\mathcal{N} = 4$ supersymmetry these operators transform. We hope to address this question in the future.

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
