# Peer review of "The Infrared Physics of Bad Theories"

_SciPost Physics, doi:SciPost Phys. 3, 024 (2017)_

## Round 2 · Referee Report · Anonymous (Referee 2) · 2017-8-24

Strengths

See report

Weaknesses

See report

Report

The paper deals with the infrared physics realized on the moduli space of $3D$ $\mathcal{N}=4$ supersymmetric gauge theory, with gauge group $U\left(N_{c}\right)$ and $N_{f}$ fundamental flavors, in a part of the range of $N_{f}$ where the theory is known as \emph{bad} $N_{f}\le2N_{c}-2$ or \emph{ugly} $N_{f}=2N_{c}-1$. The authors use recent results on the quantum corrected structure of the moduli space to describe the physics in the vicinity of its singularities. These are associated with interacting superconformal fixed points. The physics at each point is that of an SCFT associated with a \emph{good} theory ($\tilde{N}_{f}\ge2\tilde{N}_{c}$) together with a decoupled free sector. The main results are a computation of the maximal rank of the interacting part; a description of the decoupled sector; and the identification of a special \emph{symmetric vacuum} where all global symmetries of the theory remain unbroken.

A previous attempt at elucidating the physics for \emph{bad} theories conjectured a Seiberg-like infrared duality between the bad theory and a good theory with $\tilde{N}_{c}=N_{f}-N_{c}$, $\tilde{N}_{f}=N_{f}$ flavors, and a set of $2N_{c}-N_{f}$ free twisted hypermultiplets. Inclusion of the twisted hypermultiplets was motivated by the appearance, in the bad theory, of unitary violating monopole operators, in analogy with a previous analysis of \emph{ugly} theories. The authors show, however, that the proposed duality is only a good effective description in the neighborhood of the \emph{symmetric vacuum}. Higher rank singularities, not described by the proposed dual, occur elsewhere on the moduli space. The authors use the existence and properties of the \emph{symmetric vacuum} to nevertheless explain the results coming from supersymmetric localization, which motivated the proposed duality.

The paper is well written and the analysis clearly supports the conclusions. I recommend its publication in SciPost Physics.

Requested changes

None.

  • validity: top
  • significance: good
  • originality: good
  • clarity: top
  • formatting: excellent
  • grammar: excellent

Author:  Benjamin Assel  on 2017-09-05  [id 165]

(in reply to Report 2 on 2017-08-24)
Category:
remark

We thank the referee for the time spent reading our paper and for the nice comments.

---

## Round 2 · Referee Report · Anonymous (Referee 1) · 2017-8-24

Strengths

  1. The introduction is very well written: Motivation is clear and the background physics are summarized very well.
  2. The paper deals with a class of theories which has not been studied much before.
  3. The logic in the main text is clear and easy to understand, which is well supported by explicit examples presented in appendices.

Weaknesses

  1. The paper lacks a bit of novelty: The authors mainly use the tools that have been developed in [15-17].

Report

The authors discuss the infrared physics of bad 3d N=4 theories, which have not drawn much attention so far. Using the algebraic description of the Coulomb branch, they derive the global and local geometry of the moduli space of vacua of 3d N=4 theories. While the result reproduces previously known low energy physics for good and ugly theories, they find a disagreement for bad theories: They disprove a Seiberg duality for the bad theories proposed in [8], by showing that the global structures of the two dual moduli spaces do not match each other. They also explain how the two theories which are not dual can show an agreement in three-sphere partition functions.

The logics presented in the paper is clear and the physics behind various results are very well explained. Although they mainly use the descriptions which have been already developed in [15-17], the observation they have is original and somewhat unexpected. Therefore I highly recommend this paper for the publication after clarifying a minor issue below.

Requested changes

  1. In section 2.3 and so on, it is claimed that ALL the singularities in the Coulomb branch can be understood as degenerating locus of the Jacobian matrix (2.23), which does not sound obvious. For example, how does the massless W-boson singularity fit into this picture?

  • validity: high
  • significance: good
  • originality: low
  • clarity: high
  • formatting: excellent
  • grammar: excellent

Author:  Benjamin Assel  on 2017-09-05  [id 166]

(in reply to Report 1 on 2017-08-24)
Category:
answer to question

We thank the referee for the time spent reading our paper and for the nice comments. Below we reply to the point raised in the report.

The singular subvariety of an algebraic variety is by definition the subspace where the Jacobian matrix degenerates and we have pointed to the fact that at singular points there are massless fields, which comprise massless hypermultiplets and possibly massless W-boson multiplets. The low-energy effective theory on the codimension $r$ singular locus (see Equation (2.32)) accounts for all the massless degrees of freedom, including the massless $U(r)$ W-bosons. This corresponds to having r vanishing triples, $(\varphi_a,u^\pm_a)=(0,0,0)$. On the other hand, smooth points on the Coulomb branch can be described in terms of abelian variables with the triples $(\varphi_a,u^\pm_a)$ all different (and non-zero), leading to massive W-bosons.
Physically, there is a ``repulsive force" between the scalars $\varphi_a$, which prevents the W-boson to be massless, unless enough matter fields are also massless. For $N_f=0,1,2$ the W-bosons never become massless on the Coulomb branch.

We hope that this addresses the question in the report.

---

## Editorial Decision

published